# ConFIG: Towards Conflict-free Training of Physics Informed Neural Networks

**Qiang Liu[1], Mengyu Chu[2] & Nils Thuerey[1]**

School of Computation, Information and Technology [1]
Technical University of Munich
Garching, DE 85748
{qiang7.liu, nils.thuerey}@tum.de

SKL of General AI [2]
Peking University
Beijing, CN 100871
mchu@pku.edu.cn

## Abstract

The loss functions of many learning problems contain multiple additive terms that can disagree and yield conflicting update directions. For Physics-Informed Neural Networks (PINNs), loss terms on initial/boundary conditions and physics equations are particularly interesting as they are well-established as highly difficult tasks. To improve learning the challenging multi-objective task posed by PINNs, we propose the ConFIG method, which provides conflict-free updates by ensuring a positive dot product between the final update and each loss-specific gradient. It also maintains consistent optimization rates for all loss terms and dynamically adjusts gradient magnitudes based on conflict levels. We additionally leverage momentum to accelerate optimizations by alternating the back-propagation of different loss terms. We provide a mathematical proof showing the convergence of the ConFIG method, and it is evaluated across a range of challenging PINN scenarios. ConFIG consistently shows superior performance and runtime compared to baseline methods. We also test the proposed method in a classic multi-task benchmark, where the ConFIG method likewise exhibits a highly promising performance. Source code is available at `https://tum-pbs.github.io/ConFIG`

## 1 Introduction

Efficiently solving partial differential equations (PDEs) is crucial for various scientific fields such as fluid dynamics, electromagnetics, and financial mathematics. However, the nonlinear and high-dimensional PDEs often present significant challenges to the stability and convergence of traditional numerical methods. With the advent of deep learning, there is a strongly growing interest in using this technology to solve PDEs (Han et al., 2018; Beck et al., 2023). In this context, Physics Informed Neural Networks (PINNs) (Raissi et al., 2019; Cuomo et al., 2022) leverage networks as a continuous and differentiable ansatz for the underlying physics.

PINNs use auto-differentiation of coordinate-based neural networks, i.e., implicit neural representation (INR), to approximate PDE derivatives. The residuals of the PDEs, along with boundary and initial conditions, are treated as loss terms and penalized during training to achieve a physically plausible solution. Despite their widespread use, training PINNs is a well-recognized challenge (Cuomo et al., 2022; Lino et al., 2023; Wang et al., 2021; Krishnapriyan et al., 2021; Wang et al., 2022) due to several possible factors like unbalanced back-propagated gradients from numerical stiffness (Wang et al., 2021), different convergence rates among loss terms (Wang et al., 2022), PDE-based soft constraints (Krishnapriyan et al., 2021), poor initialization (Wong et al., 2024), and suboptimal sampling strategies (Daw et al., 2023). Traditional methods to improve PINN training typically involve adjusting the weights for PDE residuals and loss terms for initial/boundary conditions (Liu & Wang, 2021; McClenny & Braga-Neto, 2023; Son et al., 2023). However, although these methods claim to have better solution accuracy, there is currently no consensus on the optimal weighting strategy.

Meanwhile, methods manipulating the gradient of each loss term have become popular in Multi-Task Learning (MTL) and Continual Learning (CL) (Riemer et al., 2019; Farajtabar et al., 2020; Yu et al., 2020) to address *conflicts* between loss-specific gradients that induce negative transfer (Long et al., 2017) and catastrophic forgetting (Kirkpatrick et al., 2017). Unlike weighting strategies that

adjust the final update direction solely by modifying each loss-specific gradient's magnitude, these methods take greater flexibility and often alter the directions of loss-specific gradients. We notice that similar conflicts between gradients also arise in the training of PINNs. First, the gradient magnitude of PDE residuals typically surpasses that of initial/boundary conditions (Wang et al., 2021), causing the final update gradient to lean heavily towards the residual term. Additionally, PDE residuals often have many local minima due to the infinite number of PDE solutions without prescribed initial/boundary conditions (Daw et al., 2023). Consequently, when the optimization process approaches local minima of the PDE residual, the combined update conflicts with the gradient of initial/boundary conditions, severely impeding learning progress.

To illustrate gradient conflicts in PINNs, we show a toy case in Fig. 1 where $\mathcal{L}_1$ has smaller gradients and $\mathcal{L}_2$ has larger gradients but multiple minima. Training with $\mathcal{L}_1 + \lambda\mathcal{L}_2$ using suboptimal $\lambda$ values (e.g., $\lambda = 1$) leads to gradient conflicts and the convergence to a local minima of $\mathcal{L}_2$.

In the following, we propose the **Con**flict-**F**ree **I**nverse **G**radients (ConFIG) method to mitigate the conflicts of loss-specific gradients during PINNs training. Our approach provides an update gradient $\boldsymbol{g}_{\text{ConFIG}}$ that reduces all loss terms to prevent the optimization from being stuck in the local

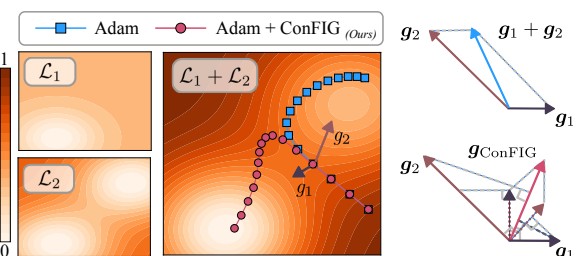

Figure 1: Visualization of toy example showing the conflict between different losses during optimization.

minima of a specific loss term. With the ConFIG method, the optimization in Fig. 1 converges to the shared minimum for both losses. Our method is provably convergent, and characterized by the following properties:

- The final update direction does not conflict with any loss-specific gradients.
- The projection length of the final gradient on each loss-specific gradient is uniform, ensuring that all loss terms are optimized at the same rate.
- The length of the final gradient is adaptively scaled based on the degree of conflict between loss-specific gradients. This prevents the optimization from stalling in the local minima of any specific loss term in high-conflict scenarios.

In addition, we introduce a momentum-based approach to expedite optimization. By leveraging momentum, we eliminate the need to compute all gradients via backpropagation at each training iteration. In contrast, we evaluate alternating loss-specific gradients, which significantly decreases the computational cost and leads to more accurate solutions within a given computational budget.

## 2   RELATED WORK

**Weighting Strategies for PINNs' Training**   Some intuitive weighting strategies come from the penalty method of constrained optimization problems, where higher weights are assigned to less-optimized losses (Liu & Wang, 2021; Son et al., 2023). McClenny & Braga-Neto (2023) extended this idea by directly setting weights to sample points with soft attention masks. On the contrary, Bischof & Kraus (2021) advocated equalizing the reduction rates of all loss terms as it guides training toward Pareto optimal solutions from an MTL perspective. Meanwhile, Wang et al. (2021) addressed numerical stiffness issues by determining the weight according to the magnitude of each loss-specific gradient. Wang et al. (2022) explored training procedures via the Neural Tangent Kernel (NTK) perspective, adaptively setting weights based on NTK eigenvalues. Xiang et al. (2022) instead employed Gaussian probabilistic models for uncertainty quantification, estimating weights through maximum likelihood estimation. Others have pursued specialized approaches. E.g., van der Meer et al. (2022) derived optimal weight parameters for specific problems and devised heuristics for general problems. de Wolff et al. (2022) use an evolutionary multi-objective algorithm to find the trade-offs between individual loss terms in PINNs training. Shin et al. (2020) conducted convergence analyses for PINNs solving linear second-order elliptic and parabolic type PDEs, employing Lipschitz regularized loss. In contrast to adaptive methods, Wight & Zhao (2021) proposed

a fixed weighting strategy for phase field models. They artificially elevated weights for the loss term associated with initial conditions.

**Gradient Improvement Strategies in MTL and CL**   Gradient improvement strategies in MTL and CL focus on understanding and resolving conflicts between loss-specific gradients. Riemer et al. (2019) and Du et al. (2018) proposed using the dot product (cosine similarity) of two gradient vectors to assess whether updates will conflict with each other. Riemer et al. (2019) introduced an additional term in the loss to modify the gradient direction, maximizing the dot product of the gradients. Conversely,Du et al. (2018) chose to discard gradients of auxiliary tasks if they conflict with the main task. Another intuitive method to resolve conflicts is orthogonal projection. Farajtabar et al. (2020) proposed the Orthogonal Gradient Descent (OGD) method for continual learning, projecting gradients from new tasks onto a subspace orthogonal to the previous task. Chaudhry et al. (2020) proposed learning tasks in low-rank vector subspaces that are kept orthogonal to minimize interference. Yu et al. (2020) introduced the PCGrad method that projects a task's gradient onto the orthogonal plane of other gradients. Liu et al. (2021b) designed the IMTL-G method to ensure that the final gradient has the same projection length on all other vectors. Dong et al. (2022) utilized the Singular Value Decomposition (SVD) on gradient vectors to obtain an orthogonal basis. Javaloy & Valera (2022) introduces a method that jointly homogenizes gradient magnitudes and directions for MTL. Quinton & Rey (2024) propose a Jacobian descent method with an aggregation step, which constrains the update gradient into the positive cone in the parameter space to avoid conflicts during training. Few studies in the PINN community have utilized gradient-based improvement strategies. Zhou et al. (2023) introduced the PCGrad method to PINN training for reliability assessment of multi-state systems. Yao et al. (2023) developed a MultiAdam method where they apply Adam optimizer for each loss term separately. Concurrent work from Hwang & Lim (2024) proposed a dual cone gradient descent method to mitigate gradient conflict when training PINNs.

## 3   METHOD

### 3.1   CONFLICT-FREE INVERSE GRADIENTS METHOD

Generically, we consider an optimization procedure with a set of $m$ individual loss functions, i.e., $\{\mathcal{L}_1, \mathcal{L}_2, \cdots, \mathcal{L}_m\}$. Let $\{\boldsymbol{g}_1, \boldsymbol{g}_2, \cdots, \boldsymbol{g}_m\}$ denote the individual gradients corresponding to each of the loss functions. A gradient-descent step with gradient $\boldsymbol{g}_c$ will conflict with the decrease of $\mathcal{L}_i$ if $\boldsymbol{g}_i^\top \boldsymbol{g}_c$ is *negative* (Riemer et al., 2019; Du et al., 2018). Thus, to ensure that all losses are decreasing simultaneously along $\boldsymbol{g}_c$, all $m$ components of $[\boldsymbol{g}_1, \boldsymbol{g}_2, \cdots, \boldsymbol{g}_m]^\top \boldsymbol{g}_c$ should be *positive*. This condition is fulfilled by setting $\boldsymbol{g}_c = [\boldsymbol{g}_1, \boldsymbol{g}_2, \cdots, \boldsymbol{g}_m]^{-\top} \boldsymbol{w}$, where $\boldsymbol{w} = [w_1, w_2, \cdots, w_m]$ is a vector with $m$ positive components and $M^{-\top}$ is the pseudoinverse of the transposed matrix $M^\top$.

Although a positive $\boldsymbol{w}$ vector guarantees a conflict-free update direction for all losses, the specific value of $w_i$ further influences the exact direction of $\boldsymbol{g}_c$. To facilitate determining $\boldsymbol{w}$, we reformulate $\boldsymbol{g}_c$ as $\boldsymbol{g}_c = k[\mathcal{U}(\boldsymbol{g}_1), \mathcal{U}(\boldsymbol{g}_2), \cdots, \mathcal{U}(\boldsymbol{g}_m)]^{-\top} \hat{\boldsymbol{w}}$, where $\mathcal{U}(\boldsymbol{g}_i) = \boldsymbol{g}_i/(|\boldsymbol{g}_i| + \varepsilon)$ is a normalization operator and $k > 0$. Now, $k$ controls the length of $\boldsymbol{g}_c$ and the ratio of $\hat{\boldsymbol{w}}$'s components corresponds to the ratio of $\boldsymbol{g}_c$'s projections onto each loss-specific $\boldsymbol{g}_i$, i.e., $|\boldsymbol{g}_c|\mathcal{S}_c(\boldsymbol{g}, \boldsymbol{g}_i)$, where $\mathcal{S}_c(\boldsymbol{g}_i, \boldsymbol{g}_j) = \boldsymbol{g}_i^\top \boldsymbol{g}_j/(|\boldsymbol{g}_i||\boldsymbol{g}_j| + \varepsilon)$ is the operator for cosine similarity:

$$\frac{|\boldsymbol{g}_c|\mathcal{S}_c(\boldsymbol{g}_c, \boldsymbol{g}_i)}{|\boldsymbol{g}_c|\mathcal{S}_c(\boldsymbol{g}_c, \boldsymbol{g}_j)} = \frac{\mathcal{S}_c(\boldsymbol{g}_c, \boldsymbol{g}_i)}{\mathcal{S}_c(\boldsymbol{g}_c, \boldsymbol{g}_j)} = \frac{\mathcal{S}_c(\boldsymbol{g}_c, k\mathcal{U}(\boldsymbol{g}_i))}{\mathcal{S}_c(\boldsymbol{g}_c, k\mathcal{U}(\boldsymbol{g}_j))} = \frac{[k\mathcal{U}(\boldsymbol{g}_i)]^\top \boldsymbol{g}_c}{[k\mathcal{U}(\boldsymbol{g}_j)]^\top \boldsymbol{g}_c} = \frac{\hat{w}_i}{\hat{w}_j} \quad \forall i, j \in [1, m]. \quad (1)$$

We call $\hat{\boldsymbol{w}}$ the *direction weight*. The projection length of $\boldsymbol{g}_c$ on each loss-specific gradient serves as an effective "learning rate" for each loss. Here, we choose $\hat{w}_i = \hat{w}_j \, \forall i, j \in [1, m]$ to ensure a uniform decrease rate of all losses, as it was shown to yield a weak form of Pareto optimality for multi-task learning (Bischof & Kraus, 2021).

Meanwhile, we introduce an adaptive strategy for the length of $\boldsymbol{g}_c$ rather than directly setting a fixed value of $k$. We notice that the length of $\boldsymbol{g}_c$ should increase when all loss-specific gradients point nearly in the same direction since it indicates a favorable direction for optimization. Conversely, when loss-specific gradients are close to opposing each other, the magnitude of $\boldsymbol{g}_c$ should decrease. We realize this by rescaling the length of $\boldsymbol{g}_c$ to the sum of the projection lengths of each loss-specific gradient on it, i.e., $|\boldsymbol{g}_c| = \sum_{i=1}^m |\boldsymbol{g}_i|\mathcal{S}_c(\boldsymbol{g}_i, \boldsymbol{g}_c)$.

The procedures above are summarized in the ***Conflict-Free Inverse Gradients (ConFIG)*** operator $G$ and we correspondingly denote the final update gradient $\boldsymbol{g}_c$ with $\boldsymbol{g}_{\text{ConFIG}}$:

$$\boldsymbol{g}_{\text{ConFIG}} = \mathcal{G}(\boldsymbol{g}_1, \boldsymbol{g}_1, \cdots, \boldsymbol{g}_m) := \left( \sum_{i=1}^{m} \boldsymbol{g}_i^\top \boldsymbol{g}_u \right) \boldsymbol{g}_u, \tag{2}$$

$$\boldsymbol{g}_u = \mathcal{U}\left[ [\mathcal{U}(\boldsymbol{g}_1), \mathcal{U}(\boldsymbol{g}_2), \cdots, \mathcal{U}(\boldsymbol{g}_m)]^{-\top} \mathbf{1}_m \right]. \tag{3}$$

Here, $\mathbf{1}_m$ is a unit vector with $m$ components. A mathematical proof of ConFIG's convergence in convex and non-convex landscapes can be found in Appendix A.1. The ConFIG method utilizes the pseudoinverse of the gradient matrix to obtain a conflict-free direction. In Appendix A.3, we prove that such an inverse operation is always feasible as long as the dimension of parameter space is larger than the number of losses. Besides, while calculating the pseudoinverse numerically could involve additional computational cost, it is not significant compared to the cost of back-propagation for each loss term. A detailed breakdown of the computational cost can be found in Appendix A.6.

## 3.2 Two Term Losses and Positioning w.r.t. Existing Approaches

For the special case of only two loss terms, there is an equivalent form of ConFIG that does not require a pseudoinverse:

$$\mathcal{G}(\boldsymbol{g}_1, \boldsymbol{g}_2) = (\boldsymbol{g}_1^\top \boldsymbol{g}_v + \boldsymbol{g}_2^\top \boldsymbol{g}_v) \boldsymbol{g}_v \tag{4}$$

$$\boldsymbol{g}_v = \mathcal{U}\left[ \mathcal{U}(\mathcal{O}(\boldsymbol{g}_1, \boldsymbol{g}_2)) + \mathcal{U}(\mathcal{O}(\boldsymbol{g}_2, \boldsymbol{g}_1)) \right] \tag{5}$$

where $\mathcal{O}(\boldsymbol{g}_1, \boldsymbol{g}_2) = \boldsymbol{g}_2 - \frac{\boldsymbol{g}_1^\top \boldsymbol{g}_2}{|\boldsymbol{g}_1|^2} \boldsymbol{g}_1$ is the orthogonality operator. It returns a vector orthogonal to $\boldsymbol{g}_1$ from the plane spanned by $\boldsymbol{g}_1$ and $\boldsymbol{g}_2$. The proof of equivalence is shown in Appendix A.4.

PCGrad (Yu et al., 2020) and IMTL-G (Liu et al., 2021b) methods from multi-task learning studies also have a similar simplified form for the two-loss scenario. PCGrad projects loss-specific gradients onto the normal plane of others if their cosine similarity is negative, while IMTL-G rescales loss-specific gradients to equalize the final gradient's projection length on each loss-specific gradient, as illustrated in Fig. 2a and 2b. Our ConFIG method in Fig. 2c employs orthogonal components of each loss-specific gradient, akin to PCGrad. Meanwhile, it also ensures the same decrease rate of all losses, similar to IMTL-G. As a result, these three methods share an identical update direction but a different update magnitude in the two-loss scenario. This provides a valuable opportunity to evaluate our adaptive magnitude strategy: for two losses, the PCGrad and IMTL-G methods can be viewed as ConFIG variants with different magnitude rescaling strategies.

The above similarity between these three methods only holds for the two-loss scenario. With more losses involved, the differences between them are more evident. In fact, the inverse operation in Eq. 3 makes the ConFIG approach the only method that maintains a conflict-free direction when the number of loss terms exceeds two. Detailed discussion can be found in A.2

## 3.3 ConFIG with Momentum Acceleration

Gradient-based methods like PCGrad, IMTL-G, and the proposed ConFIG method require separate backpropagation steps to compute the gradient for each loss term. In contrast, conventional weighting strategies only require a single backpropagation for the total loss. Thus, gradient-based methods

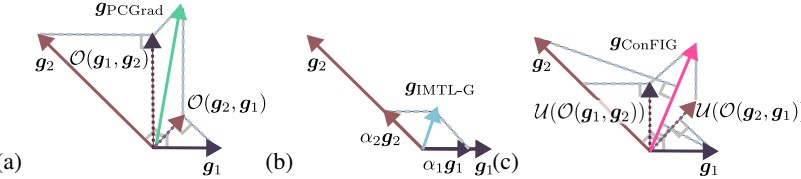

Figure 2: Sketch of PCGrad (a), IMTL-G (b), and our ConFIG (c) method with two loss terms. The PCGrad method directly sums two orthogonal components, and the IMTL-G method rescales the two vectors to the same magnitude. Our ConFIG method sums the unit vector of the orthogonal components and adjusts its magnitude with the projection length of each loss-specific gradient.

---

**Algorithm 1** M-ConFIG

---

**Require:** $\theta_0$ (network weights), $\gamma$ (learning rate), $\beta_1$, $\beta_2$, $\epsilon$ (Adam coefficient), $\boldsymbol{m}_0$ (Pseudo first momentum),
  $[\boldsymbol{m}_{\boldsymbol{g}_1,0}, \boldsymbol{m}_{\boldsymbol{g}_1,0}, \cdots \boldsymbol{m}_{\boldsymbol{g}_m,0}] \leftarrow [\boldsymbol{0}, \boldsymbol{0}, \cdots, \boldsymbol{0}]$ (First momentum), $\boldsymbol{v}_0 \leftarrow \boldsymbol{0}$ (Second momentum),
  $[t_{\boldsymbol{g}_1}, t_{\boldsymbol{g}_2}, \cdots, t_{\boldsymbol{g}_m}] \leftarrow [0, 0, \cdots, 0]$, All operations on vectors are element-wise except $\mathcal{G}$.
  **for** $t \leftarrow 1$ to $\cdots$ **do**
    $i = t\%m + 1$
    $t_{\boldsymbol{g}_i} \leftarrow t_{\boldsymbol{g}_i} + 1$
    $\boldsymbol{m}_{\boldsymbol{g}_i, t_{\boldsymbol{g}_i}} \leftarrow \beta_1 \boldsymbol{m}_{\boldsymbol{g}_i, t_{\boldsymbol{g}_i}-1} + (1-\beta_1)\nabla_{\theta_{t-1}}\mathcal{L}_i$ ▷ Update the first momentum of $\boldsymbol{g}_i$
    $[\hat{\boldsymbol{m}}_{\boldsymbol{g}_1}, \hat{\boldsymbol{m}}_{\boldsymbol{g}_2}, \cdots, \hat{\boldsymbol{m}}_{\boldsymbol{g}_m}] \leftarrow [\frac{\boldsymbol{m}_{\boldsymbol{g}_1,t_{\boldsymbol{g}_1}}}{1-\beta_1^{t_{\boldsymbol{g}_1}}}, \frac{\boldsymbol{m}_{\boldsymbol{g}_2,t_{\boldsymbol{g}_2}}}{1-\beta_1^{t_{\boldsymbol{g}_2}}}, \cdots, \frac{\boldsymbol{m}_{\boldsymbol{g}_m,t_{\boldsymbol{g}_m}}}{1-\beta_1^{t_{\boldsymbol{g}_m}}}]$ ▷ Bias corrections for first momentum terms
    $\hat{\boldsymbol{m}}_g \leftarrow \mathcal{G}(\hat{\boldsymbol{m}}_{\boldsymbol{g}_1}, \hat{\boldsymbol{m}}_{\boldsymbol{g}_2}, \cdots, \hat{\boldsymbol{m}}_{\boldsymbol{g}_m})$ ▷ ConFIG update of momentums
    $\boldsymbol{g}_c \leftarrow [\hat{\boldsymbol{m}}_g(1-\beta_1^t) - \beta_1 \boldsymbol{m}_{t-1}]/(1-\beta_1)$ ▷ Obtain the estimated gradient
    $\boldsymbol{m}_t \leftarrow \beta_1 \boldsymbol{m}_{t-1} + (1-\beta_1)\boldsymbol{g}_c$ ▷ Update the pseudo first momentum
    $\boldsymbol{v}_t \leftarrow \beta_2 \boldsymbol{v}_{t-1} + (1-\beta_2)\boldsymbol{g}_c^2$ ▷ Update the second momentum
    $\hat{\boldsymbol{v}} \leftarrow \boldsymbol{v}_t/(1-\beta_2^t)$ ▷ Bias correction for the second momentum
    $\theta_i \leftarrow \theta_{t-1} - \gamma\hat{\boldsymbol{m}}_g/(\sqrt{\hat{\boldsymbol{v}}_g} + \epsilon)$ ▷ Update weights of the neural network
  **end for**

---

are usually $r = \sum_i^m \mathcal{T}_b(\mathcal{L}_i)/\mathcal{T}_b\left(\sum_i^m \mathcal{L}_i\right)$ times more computationally expensive than weighting methods, where $\mathcal{T}_b$ is the computational cost of backpropagation. To address this issue, we introduce an accelerated momentum-based variant of ConFIG: *M-ConFIG*. Our core idea is to leverage the momentum of the gradient for the ConFIG operation and update momentum in an alternating fashion to avoid backpropagating all losses in a single step. In each iteration, only a single momentum is updated with its corresponding gradient, while the others are carried over from previous steps. This reduces the computational cost of the M-ConFIG method to $1/m$ of the ConFIG method.

Algorithm 1 details the entire procedure of M-ConFIG. It aligns with the fundamental principles of the Adam algorithm, where the first momentum averages the local gradient, and the second momentum adjusts the magnitude of each parameter. In this approach, we calculate only a single second momentum using an estimated gradient based on the output of the ConFIG operation. An alternative could involve calculating the second momentum for every loss term, similar to the MultiAdam method (Yao et al., 2023). However, we found this strategy to be inaccurate and numerically unstable compared to Algorithm 1. A detailed discussion and comparison can be found in the Appendix 1.

Surprisingly, M-ConFIG not only catches up with the training speed of regular weighting strategies but typically yields an even lower average computational cost per iteration. This stems from the fact that backpropagating a sub-loss $\mathcal{L}_i$ is usually faster than backpropagating the total loss $\sum_i^m \mathcal{L}_i$. Thus, $r$ is usually smaller than $m$ and $r/m < 1$. This is especially obvious for PINN training, where a reduced number of sampling points are used for boundary/initial terms and are faster to evaluate than the residual term. In our experiments, we observed an average value of $r = 1.67$ and a speed-up of $1.67/3 \approx 0.56$ per iteration for PINNs trained with three losses using M-ConFIG.

## 4 EXPERIMENTS

In this section, we employ the proposed methods for training PINNs on several challenging PDEs. We also explore the application of our method on a classical Multi-Task Learning (MTL) benchmark

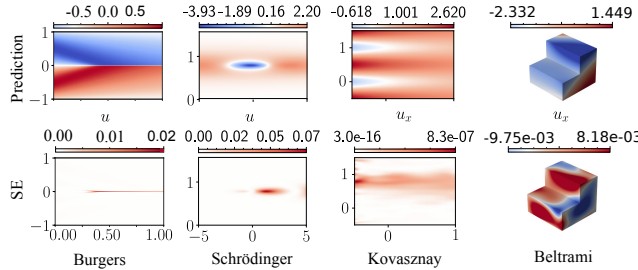

Figure 3: Examples of PINN predictions and squared error (SE) distributions on the test PDEs.

as an outlook. Unless mentioned otherwise, every result is computed via averaging three training runs initialized with different random seeds, each using the model with the best test performance during the training. For detailed numerical values and standard deviations of the result in each experiment, please refer to the Appendix A.9 and A.10. Training configurations and hyper-parameters of each experiment can be found in the Appendix A.12. An additional ablation study on training hyperparameters can be found in the Appendix A.13. It shows that the gains from ConFIG are robust w.r.t. hyperparameter changes.

## 4.1 PHYSICS INFORMED NEURAL NETWORKS

**Preliminaries of PINNs.** Consider the initial boundary value problem of a general PDE for a scalar function $u(\boldsymbol{x}, t) \colon \mathbb{R}^{d+1} \to \mathbb{R}$ given by

$$\mathcal{N}[u(\boldsymbol{x},t), \boldsymbol{x}, t] := N[u(\boldsymbol{x},t), \boldsymbol{x}, t] + f(\boldsymbol{x}, t) = 0, \quad \boldsymbol{x} \in \Omega, t \in (0, T], \tag{6}$$

$$\mathcal{B}[u(\boldsymbol{x},t), \boldsymbol{x}, t] := B[u(\boldsymbol{x},t), \boldsymbol{x}, t] + g(\boldsymbol{x}, t) = 0, \quad \boldsymbol{x} \in \partial\Omega, t \in (0, T], \tag{7}$$

$$\mathcal{I}[u(\boldsymbol{x},0), \boldsymbol{x}, 0] := u(\boldsymbol{x}, 0) + h(\boldsymbol{x}, 0) = 0, \quad \boldsymbol{x} \in \overline{\Omega}. \tag{8}$$

Here, $\Omega \subset R^d$ is a spatial domain with $\partial\Omega$ and $\overline{\Omega}$ denotes its boundary and closure, respectively. $N$ and $B$ are spatial-temporal differential operators, and $f$, $g$ and $h$ are source functions. To solve this initial boundary value problem, PINNs introduce a neural network $\hat{u}(\boldsymbol{x}, t, \boldsymbol{\theta})$ for the target function $u(\boldsymbol{x}, t)$, where $\theta$ is the weights of the neural networks. Then, spatial and temporal differential operators can be calculated efficiently with auto-differentiation of $N[\hat{u}(\boldsymbol{x}, t, \boldsymbol{\theta}), \boldsymbol{x}, t]$ and $B[\hat{u}(\boldsymbol{x}, t, \boldsymbol{\theta}), \boldsymbol{x}, t]$. Solving Eq. 6-8 turns to the training of neural networks with a loss function of

$$\mathcal{L}(\boldsymbol{\theta}) = \underbrace{\sum_{i=1}^{n_{\mathcal{N}}} \mathcal{N}[\hat{u}(\boldsymbol{x}_i, t_i, \boldsymbol{\theta}), \boldsymbol{x}_i, t_i]}_{\mathcal{L}_{\mathcal{N}}} + \underbrace{\sum_{i=1}^{n_{\mathcal{B}}} \mathcal{B}[\hat{u}(\boldsymbol{x}_i, t_i, \boldsymbol{\theta}), \boldsymbol{x}_i, t_i]}_{\mathcal{L}_{\mathcal{B}}} + \underbrace{\sum_{i=1}^{n_{\mathcal{I}}} \mathcal{I}[\hat{u}(\boldsymbol{x}_i, 0, \boldsymbol{\theta}), \boldsymbol{x}_i, 0]}_{\mathcal{L}_{\mathcal{I}}}, \tag{9}$$

where $(\boldsymbol{x}_i, t_i)$ are the spatial-temporal coordinates for the data samples in the $\mathbb{R}^{d+1}$ domain, $\mathcal{L}_{\mathcal{N}}$, $\mathcal{L}_{\mathcal{B}}$ and $\mathcal{L}_{\mathcal{I}}$ are the loss functions for the PDE residual, spatial and initial boundaries, respectively. These three loss terms give us three corresponding gradients for optimization: $\boldsymbol{g}_{\mathcal{N}}$, $\boldsymbol{g}_{\mathcal{B}}$, and $\boldsymbol{g}_{\mathcal{I}}$. In the experiments, we consider four cases with three different PDEs: 1D unsteady Burgers equation, 1D unsteady Schrödinger equation, 2D Kovasznay flow (Navier-stokes equations), and 3D unsteady Beltrami flow (Navier-stokes equations). Fig. 3 shows examples of the solution domain of each PDEs. A more detailed illustration and discussion of the PDEs and corresponding solution domains can be found in Appendix A.7 and A.8, respectively.

**Focusing on two loss terms.** As detailed above, the similarity between our ConFIG method and the PCGrad/IMTL-G methods in the two-loss scenario offers a valuable opportunity to evaluate the proposed strategy for adapting the magnitude of the update. Therefore, we begin with a two-loss scenario where we introduce a new composite gradient, $\mathcal{L}_{\mathcal{BI}}$, which aggregates the contributions from both the boundary ($\mathcal{L}_{\mathcal{B}}$) and initial ($\mathcal{L}_{\mathcal{I}}$) conditions. This setup also provides better insights into the conflicting updates between PDE residuals and other loss terms.

Fig. 4 compares the performance of PINNs trained with our ConFIG method and other existing approaches. Specifically, we compare to PCGrad (Yu et al., 2020) and the IMTL-G method (Liu

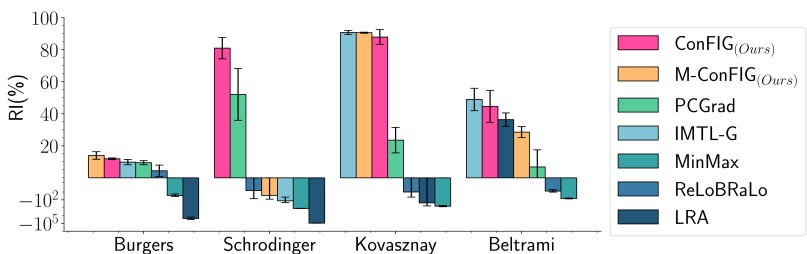

Figure 4: Relative improvements of PINNs trained with two loss terms using different methods.

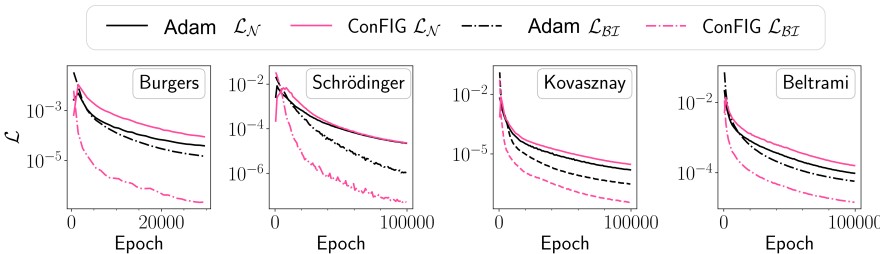

Figure 5: Training losses of PINNs trained with Adam baselines and ConFIG using two loss terms.

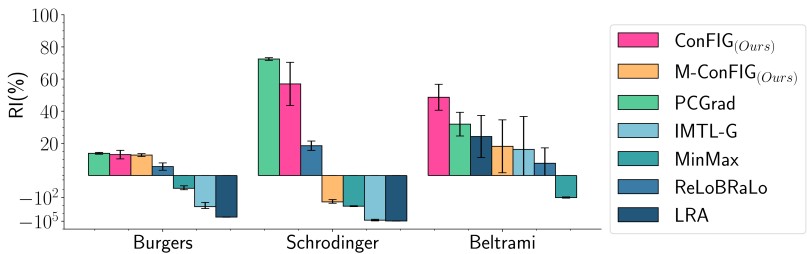

Figure 6: Relative improvements of PINNs trained with three loss terms using different methods.

et al., 2021b) from MTL studies as well as the LRA method (Wang et al., 2021), MinMax (Liu & Wang, 2021), and ReLoBRaLo (Bischof & Kraus, 2021) as established weighting methods for PINNs. The accuracy metric is the MSE between the predictions and the ground truth value on the new data points sampled in the computational domain that differ from the training data points. As the central question for all these methods is how much improvement they yield in comparison to the standard training configuration with Adam, we show the *relative improvement* (in percent) over the Adam baseline. Thus, +50 means half, -100 twice the error of Adam, respectively. (Absolute metrics are given in the Appendix A.9)

The findings reveal that only our ConFIG and PCGrad consistently outperform the Adam baseline. In addition, the ConFIG method always performs better than PCGrad. While the IMTL-G method performs slightly better than our methods in the Kovasznay and Beltrami flow cases, it performs worse than the Adam baseline in the Schrödinger case. Fig. 5 compares the training loss of our ConFIG method with the Adam baseline approach. The results indicate that ConFIG successfully mitigates the training bias towards the PDE residual term. It achieves an improved overall test performance by significantly decreasing the boundary/initial loss while sacrificing the PDE residual loss slightly. This indicates that ConFIG succeeds in finding one of the many minima of the residual loss that better adheres to the boundary conditions, i.e., it finds a better overall solution for the PDE.

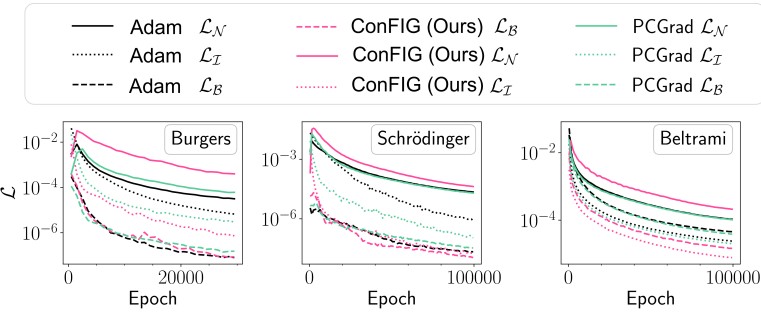

Figure 7: Training losses of the Adam baseline, PCGrad, and ConFIG with three loss terms.

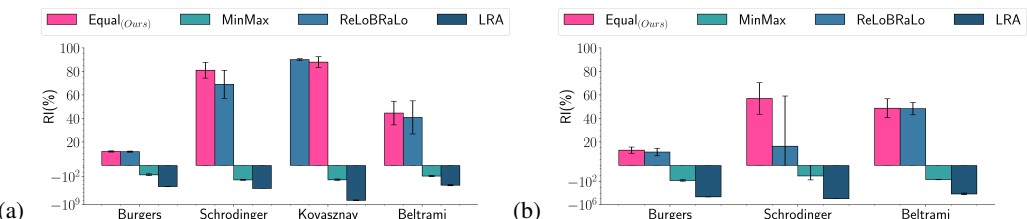

Figure 8: Relative improvements of the ConFIG method with different direction weights. (a) Two-loss scenario. (b) Three-loss scenario.

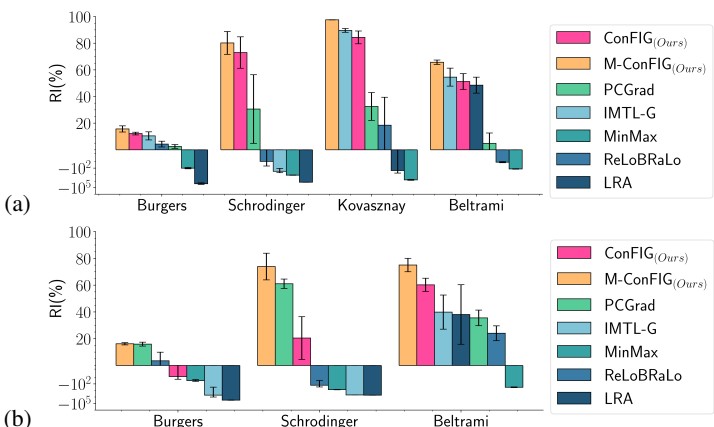

Figure 9: Relative improvements of the PINNs trained with different methods with the same wall time. (a) Two-loss scenario. (b) Three-loss scenario.

**Scenarios with three loss terms.** Fig. 6 compares the performance for all three loss terms. While the general trend persists, with ConFIG and PCGrad outperforming the other methods, these cases highlight interesting differences between these two methods. As PCGrad performs better for the Burgers and Schrödinger case, while ConFIG is better for the Beltrami flow, we analyze the training losses of both methods in comparison to Adam in Fig. 7. In the Burgers and Schrödinger equation experiment, ConFIG notably reduces the loss associated with initial conditions while the loss of boundary conditions exhibits negligible change. Similarly, for PCGrad, minimal changes are observed in boundary conditions. Furthermore, owing to PCGrad's bias towards optimizing in the direction of larger gradient magnitudes, i.e., the direction of the PDE residual, the reduction in its PDE residual term is modest, leading to a comparatively slower decrease in the loss of initial conditions compared to ConFIG. In the case of the Beltrami flow, our ConFIG method effectively reduces both boundary and initial losses, whereas the PCGrad method slightly improves boundary and initial losses. These results underscore the intricacies of PINN training, where PDE residual terms also play a pivotal role in determining the final test performance. In scenarios where the PDE residual does not significantly conflict with one of the terms, an increase in the PDE residual ultimately reduces the benefits from the improvement on the boundary/initial conditions, resulting in a better performance for PCGrad in the Burgers and Schrödinger scenario.

**Adjusting direction weights.** In our ConFIG method, we set equal components for the direction weights $\hat{w} = \mathbf{1}_m$ to ensure a uniform decrease rate across all loss terms. In the following experiments, we use the different weighting methods discussed above to calculate the components of $\hat{w}$ and compare the results with our ConFIG method. As demonstrated in Fig. 8, only our default strategy (equal weights) and the ReLoBRaLo weights get better performance than the Adam baseline. Moreover, our equal setup consistently outperforms the ReLoBRaLo method, except for a slight inferiority in the Kovasznay flow scenario with two losses. These results further validate that the equal weighting strategy is a good choice.

**Evaluation of Runtime Performance**   While the evaluations in Fig. 4 and 6 have focused on accuracy after a given number of training epochs, a potentially more important aspect for practical applications is the accuracy per runtime. This is where the M-ConFIG method can show its full potential, as its slight approximations of the update direction come with a substantial reduction in terms of computational resources (we quantify the gains per iteration in A.6). Fig. 9 compares the test MSE for M-ConFIG and other methods with a constant budget in terms of wall-clock time. Here, M-ConFIG outperforms all other methods, even PCGrad and the regular ConFIG method which yielded a better per-iteration accuracy above. This is apparent for all cases under consideration.

To shed more light on its behavior, Fig. 10 shows the test MSE of Adam, ConFIG, and M-ConFig for the most challenging scenario, the 3D unsteady Beltrami flow, with an extended training run. It shows that the advantage of M-ConFIG does not just stem from a quick late or early decrease but rather is the result of a consistently improved convergence throughout the full training. This graph additionally highlights that the regular ConFIG method, i.e., without momentum, still outperforms Adam despite its higher computational cost for each iteration.

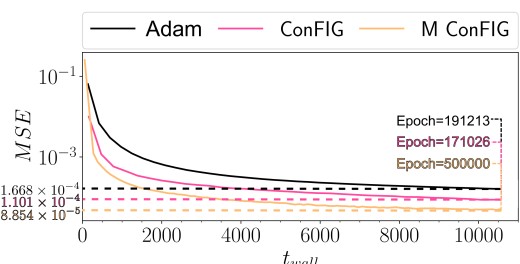

Figure 10: Test MSE of Adam baseline, ConFIG, and M-ConFIG as functions of wall time.

**Additional Tests on Challenging PDEs**   Recently, several benchmark problems have been proposed as challenging tasks for training PINNs (Hao et al., 2023). We have also tested our ConFIG and M-ConFIG methods together with other methods on these challenging tasks. The results are summarized in Appendix. A.11. They show that, although our methods do not resolve all inherent difficulties of PINN training, our methods still consistently outperform the other methods. These results shows the general appeal of the proposed methods for challenging, and high-dimensional training scenarios.

## 4.2   MULTI-TASK LEARNING

While PINNs represent a highly challenging and relevant case, we also evaluate our ConFIG method for traditional Multi-Task Learning (MTL) as an outlook. We employ the widely studied CelebA dataset (Liu et al., 2015), comprising 200,000 face images annotated with 40 facial binary attributes, making it suitable for MTL with $m = 40$ loss terms. Unlike PINNs, where each sub-task (loss term) carries distinct significance, and the final test loss may not reflect individual subtask performance, the CelebA experiment allows us to use the same metric to evaluate the performance for all tasks.

We compared the performance of our ConFIG method with ten popular MTL baselines. Besides PCGrad and IMTL-G from before, we also compare to Linear scalarization baseline (LS) (Caruana, 1997), Uncertainty Weighting (UW) (Cipolla et al., 2018), Dynamic Weight Average (DWA) (Liu et al., 2019), Gradient Sign Dropout (GradDrop) (Chen et al., 2020), Conflict-Averse Gradient Descent (CAGrad) (Liu et al., 2021a), Random Loss Weighting (RLW) (Lin et al., 2022), Nash bargaining solution for MTL (Nash-MTL) (Navon et al., 2022), and Fast Adaptive Multitask Optimization (FAMO) (Liu et al., 2023). We use two metrics to measure the performance of different methods: the mean rank metric ($MR$, lower ranks being better) (Navon et al., 2022; Liu et al., 2023), represents the average rank of across all tasks. An $MR$ of 1 means consistently outperforming all others. In addition, we consider the average F1 score ($\overline{F_1}$, larger is better) to measure the average performance on all tasks.

Fig. 11 presents a partial summary of the results. Our ConFIG method or M-ConFIG method emerges as the top-performing method for both $MR$ and $\overline{F_1}$ metric. Unlike the PINN study, we update 30 momentum variables rather than a single momentum variable for the M-ConFIG method, as indicated by the subscript '30.' This adjustment is necessary because a single update step is insufficient to achieve satisfactory results in the challenging MTL experiments involving 40 tasks.

**Varying Number of Tasks**   Fig. 12 illustrates how the performance of the M-ConFIG method varies with different numbers of tasks and momentum-update steps in the CelebA MTL experiments. The performance of M-ConFIG tends to degrade as the number of tasks increases. This

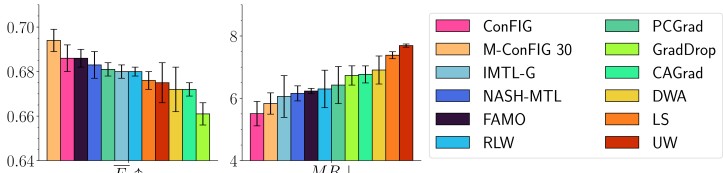

Figure 11: Test evaluation in terms of $\overline{F_1}$ and *MR* for the CelebA experiments. ConFIG and M-ConFIG 30 yield the best performance for both metrics, each of them favoring a different metric.

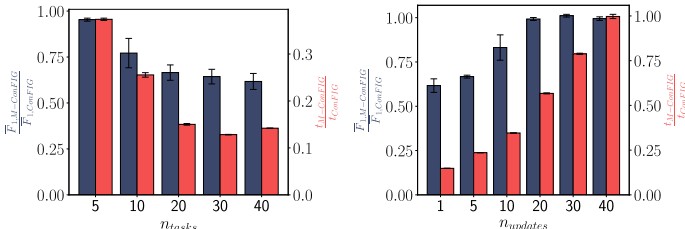

Figure 12: The relative $\overline{F_1}$ value and training time of the M-ConFIG method with a varying number of tasks and gradient update steps in the CelebA experiment.

decline is attributed to longer intervals between momentum updates for each gradient, making it difficult for the momentum to accurately capture changes in the gradient. When the number of tasks becomes sufficiently large, this performance loss outweighs the benefits of M-ConFIG's faster training speed. In our experiments, when the number of tasks reaches 10, the $\overline{F_1}$ score of the ConFIG method (0.635) already surpasses that of M-ConFIG (0.536) with the same training time. However, increasing the number of gradient update can effectively mitigate this performance degradation. In the standard 40-task MTL scenario, increasing the number of gradient update steps per iteration to 20 allows M-ConFIG to achieve a performance level comparable to ConFIG while reducing the training time to only 56% of that required by ConFIG. Notably, alternating momentum updates can sometimes even outperform the ConFIG method with the same number of training epochs, as seen in the $n_{updates} = 30$ case in the MTL experiment, as well as in the Burgers and Kovasznay cases in the two-loss PINNs experiment.

**Limitations** A potential limitation of the proposed method is the potential performance degradation of the M-ConFIG method as the number of loss terms increases. Additionally, the memory consumption required to store the momentum for each gradient grows with both the size of the parameter space and the number of loss terms. As discussed in the MTL experiment, one approach to mitigate the performance degradation is to update not just a single gradient but a stochastic subset of gradients during each iteration, albeit at the cost of an increased computational budget. Nonetheless, further investigation and improvement of the M-ConFIG method's performance in scenarios with a large number of loss terms is a promising direction for future research, aiming to broaden the applicability of the ConFIG approach. Meanwhile, we also notice that there are still many challenges in the training of PINNs, e.g., chaotic problems, that can not be addressed by eliminating the conflicts between different loss terms during the training. Future efforts, like improved network structures and imposed causality, will be required to further improve PINN training.

# 5  CONCLUSIONS

In this study, we have presented ConFIG, a method designed to alleviate training conflicts between loss terms with distinct behavior. The ConFIG method achieves conflict-free training by ensuring a positive dot product between the final gradient and each loss-specific gradient. Additionally, we introduce a momentum-based approach, replacing the gradient with alternately updated momentum for highly efficient iterations. In our experiments the proposed methods have shown superior performance compared with a wide range of existing training strategies for PINNs. The config variants even outperform SOTA methods for challenging MTL scenarios.

ACKNOWLEDGMENTS

Qiang Liu acknowledges the support from the China Scholarship Council (No.202206120036 ) for his Ph.D research at the Technical University of Munich.

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

# A APPENDIX

## A.1 CONVERGENCE PROOF FOR THE CONFIG METHOD

In this section, we will give the proof for convergence for both convex (Theorem 1) and non-convex landscapes (Theorem 2) for our ConFIG method.

**Theorem 1.** *Assume that (a) $m$ objectives $\mathcal{L}_1, \mathcal{L}_2 \cdots, \mathcal{L}_m$ are convex and differentiable; (b)The gradient $\boldsymbol{g} = \nabla_\theta \mathcal{L} = \sum_{i=1}^{m} \nabla_\theta \mathcal{L}_i$ is Lipschitz continuous with constant $L > 0$. Then, update along ConFIG direction $\boldsymbol{g}_{ConFIG}$ with step size $\gamma \leq \frac{2}{L}$ will converge to either a location where $|\boldsymbol{g}_{ConFIG}| = 0$ or the optimal solution.*

*Proof.* A Lipschitz continuous $\boldsymbol{g}$ with constant $L > 0$ gives a negative semi-definite matrix $\nabla_\theta^2 - LI$. If we do a quadratic expansion of $\mathcal{L}$ around $\mathcal{L}(\theta)$, we will get

$$
\begin{aligned}
\mathcal{L}\left(\theta^+\right) &\leq \mathcal{L}(\theta) + \nabla\mathcal{L}(\theta)^T \left(\theta^+ - \theta\right) + \frac{1}{2}\nabla^2\mathcal{L}(\theta)\left|\theta^+ - \theta\right|^2 \\
&\leq \mathcal{L}(\theta) + \boldsymbol{g}^T \left(\theta^+ - \theta\right) + \frac{1}{2}L\left|\theta^+ - \theta\right|^2.
\end{aligned}
\tag{10}
$$

The $\theta^+$ is obtained through an update as $\theta^+ = \theta - \gamma\boldsymbol{g}_{\text{ConFIG}}$, resulting in

$$
\begin{aligned}
\mathcal{L}\left(\theta^+\right) &\leq \mathcal{L}(\theta) - \gamma\boldsymbol{g}^T\boldsymbol{g}_{\text{ConFIG}} + \frac{1}{2}L\gamma^2\left|\boldsymbol{g}_{\text{ConFIG}}\right|^2 \\
&= \mathcal{L}(\theta) - \gamma(\sum_{i=1}^{m}\boldsymbol{g}_i^T)\boldsymbol{g}_{\text{ConFIG}} + \frac{1}{2}L\gamma^2\left|\boldsymbol{g}_{\text{ConFIG}}\right|^2.
\end{aligned}
\tag{11}
$$

Our $\boldsymbol{g}_{\text{ConFIG}}$ has following 2 features:

- An equal and positive projection on each loss-specific gradient: $\frac{\boldsymbol{g}_i^\top\boldsymbol{g}_{\text{ConFIG}}}{|\boldsymbol{g}_i||\boldsymbol{g}_{\text{ConFIG}}|} = \frac{\boldsymbol{g}_j^\top\boldsymbol{g}_{\text{ConFIG}}}{|\boldsymbol{g}_j||\boldsymbol{g}_{\text{ConFIG}}|} > 0$, i.e., $S_c(\boldsymbol{g}_i, \boldsymbol{g}_{\text{ConFIG}}) = S_c(\boldsymbol{g}_j, \boldsymbol{g}_{\text{ConFIG}}) := S_c > 0$;

- A magnitude equals to the sum of projection length of each loss-specific gradient on it: $|\boldsymbol{g}_{\text{ConFIG}}| = \sum_{i=1}^{m}\boldsymbol{g}_i^\top\mathcal{U}(\boldsymbol{g}_{\text{ConFIG}}) = \sum|\boldsymbol{g}_i|S_c$.

Given these two conditions, we have

$$
\begin{aligned}
(\sum_{i=1}^{m}\boldsymbol{g}_i^\top)\boldsymbol{g}_{\text{ConFIG}} &= \sum_{i=1}^{m}(\boldsymbol{g}_i^\top\boldsymbol{g}_{\text{ConFIG}}) \\
&= \sum_{i=1}^{m}(|\boldsymbol{g}_i||\boldsymbol{g}_{\text{ConFIG}}|S_c) \\
&= |\boldsymbol{g}_{\text{ConFIG}}|\sum_{i=1}^{m}(|\boldsymbol{g}_i|S_c) \\
&= |\boldsymbol{g}_{\text{ConFIG}}|^2
\end{aligned}
\tag{12}
$$

turning Eq.11 into

$$
\mathcal{L}\left(\theta^+\right) \leq \mathcal{L}(\theta) - \gamma|\boldsymbol{g}_{\text{ConFIG}}|^2 + \frac{1}{2}L\gamma^2\left|\boldsymbol{g}_{\text{ConFIG}}\right|^2
\tag{13}
$$

If $\gamma \leq \frac{2}{L}$, we have $\frac{1}{2}L\gamma^2\left|\boldsymbol{g}_{\text{ConFIG}}\right|^2 - \gamma|\boldsymbol{g}_{\text{ConFIG}}|^2 \leq 0$, which finally gives $\mathcal{L}\left(\theta^+\right) \leq \mathcal{L}\left(\theta\right)$. Note that when $\gamma \leq \frac{2}{L}$, the equality holds and only holds when $|\boldsymbol{g}_{\text{ConFIG}}| = 0$ where conflict direction doesn't exist and optimization along any direction will results in at least one of the losses increases. $\square$

**Theorem 2.** *Assume that (a) $m$ objectives $\mathcal{L}_1, \mathcal{L}_2 \cdots, \mathcal{L}_m$ are differentiable and possibly non-convex; (b)The gradient $\boldsymbol{g} = \nabla_\theta \mathcal{L} = \sum_{i=1}^{m} \nabla_\theta \mathcal{L}_i$ is Lipschitz continuous with constant $L > 0$. Then, update along ConFIG direction $\boldsymbol{g}_{ConFIG}$ with step size $\gamma \leq \frac{2}{L}$ will converge to either a location where $|\boldsymbol{g}_{ConFIG}| = 0$ or the stationary point.*

*Proof.* Using the descent lemma, an update on $\boldsymbol{g}_{\text{ConFIG}}$ with a step size $\gamma \leq \frac{2}{L}$ gives us

$$\mathcal{L}\left(\theta^{k+1}\right) \leq \mathcal{L}\left(\theta^k\right) - \frac{\gamma}{2}|\boldsymbol{g}_{\text{ConFIG}}^k|^2, \tag{14}$$

where the superscript is the index of optimization iterations.

Due to $|\boldsymbol{g}_{\text{ConFIG}}| = \sum_{i=1}^m |\boldsymbol{g}_i| S_c$, $|\boldsymbol{g}| = |\sum_{i=1}^m \boldsymbol{g}_i|$ and the triangle inequality where $\sum_{i=1}^m |\boldsymbol{g}_i| \geq |\sum_{i=1}^m \boldsymbol{g}_i|$, we have

$$
\begin{aligned}
\mathcal{L}\left(\theta^{k+1}\right) &\leq \mathcal{L}\left(\theta^k\right) - \frac{\gamma}{2}|\boldsymbol{g}_{\text{ConFIG}}^k|^2 \\
&= \mathcal{L}\left(\theta^k\right) - \frac{\gamma}{2}(\sum_{i=1}^m |\boldsymbol{g}_i^k|)^2 S_c^{k,2} \\
&\leq \mathcal{L}\left(\theta^k\right) - \frac{\gamma}{2}|\sum_{i=1}^m \boldsymbol{g}_i^k|^2 S_c^{k,2} \\
&= \mathcal{L}\left(\theta^k\right) - \frac{\gamma}{2}|\boldsymbol{g}^k|^2 S_c^{k,2}
\end{aligned} \tag{15}
$$

Summing this inequality over $k = 1, 2, \cdots, K$ results in

$$\sum_{k=1}^K |\boldsymbol{g}^k|^2 \leq \frac{2}{\gamma} \sum_{k=1}^K \frac{\mathcal{L}\left(\theta^k\right) - \mathcal{L}\left(\theta^{k+1}\right)}{S_c^{k,2}}. \tag{16}$$

By defining $\min_{1 \leq k \leq K}(S_c^k) = \alpha > 0$, we have

$$\sum_{k=1}^K |\boldsymbol{g}^k|^2 \leq \frac{2}{\gamma \alpha^2}\left[\mathcal{L}\left(\theta^0\right) - \mathcal{L}\left(\theta^{K+1}\right)\right]. \tag{17}$$

Finally, averaging Eq.17 on each side leads to

$$\min_{1 \leq k \leq K} |\boldsymbol{g}^k|^2 \leq \frac{1}{K} \sum_{k=1}^K |\boldsymbol{g}^k|^2 \leq \frac{2}{\gamma \alpha^2 K}\left[\mathcal{L}\left(\theta^0\right) - \mathcal{L}\left(\theta^{K+1}\right)\right]. \tag{18}$$

As $K \to \infty$, either the $|\boldsymbol{g}_{\text{ConFIG}}| = 0$ is obtained and the optimization stops, or the last term in the inequality goes 0, implying that the minimal gradient norm also goes to zero, i.e., a stationary point. $\square$

## A.2 COMPARISON BETWEEN CONFIG, PCGRAD AND IMTL-G

---

**Algorithm 2** PCGrad method

---

**Require:** $\theta$ (network weights), $[\mathcal{L}_1, \mathcal{L}_2, \cdots \mathcal{L}_m]$ (loss functions)
    $[\boldsymbol{g}_1, \boldsymbol{g}_2, \cdots, \boldsymbol{g}_m] \leftarrow [\nabla_\theta \mathcal{L}_1, \nabla_\theta \mathcal{L}_2, \cdots, \nabla_\theta \mathcal{L}_m]$
    $[\hat{\boldsymbol{g}}_1, \hat{\boldsymbol{g}}_2, \cdots, \hat{\boldsymbol{g}}_m] \leftarrow [\boldsymbol{g}_1, \boldsymbol{g}_2, \cdots, \boldsymbol{g}_m]$
    **for** $i \leftarrow 1$ to $m$ **do**
        **for** $j \stackrel{uniformly}{\sim} [1, 2, \cdots m]$ in random order **do**
            **if** $i \neq j$ & $\hat{\boldsymbol{g}}_i^\top \boldsymbol{g}_j < 0$ **then**
                $\hat{\boldsymbol{g}}_i = \mathcal{O}(\boldsymbol{g}_j, \hat{\boldsymbol{g}}_i)$
            **end if**
        **end for**
        $\boldsymbol{g}_{\text{PCGrad}} = \sum_{i=1}^j \hat{\boldsymbol{g}}_i$
    **end for**

---

**PCGrad method.** Algorithm 2 outlines the complete algorithm of the PCGrad method. The core idea is to use $\mathcal{O}(\boldsymbol{g}_2, \boldsymbol{g}_1)$ to replace $\boldsymbol{g}_1$ when $\boldsymbol{g}_1$ conflicts with $\boldsymbol{g}_2$ i.e., $\boldsymbol{g}_1^\top \boldsymbol{g}_2 < 0$. Since $\mathcal{O}(\boldsymbol{g}_2, \boldsymbol{g}_1)^\top \boldsymbol{g}_2 = 0$, the conflict is resolved. In the simple case of two loss terms, this results in the final gradient $\boldsymbol{g}_{\text{PCGrad}} = \mathcal{O}(\boldsymbol{g}_1, \boldsymbol{g}_2) + \mathcal{O}(\boldsymbol{g}_2, \boldsymbol{g}_1)$. This final update gradient does not conflict with

all loss-specific gradients, as $[\mathcal{O}(\boldsymbol{g}_1, \boldsymbol{g}_2) + \mathcal{O}(\boldsymbol{g}_2, \boldsymbol{g}_1)]^\top \boldsymbol{g}_1 = \mathcal{O}(\boldsymbol{g}_2, \boldsymbol{g}_1)^\top \boldsymbol{g}_1 = |\boldsymbol{g}_1|^2 - \frac{(\boldsymbol{g}_1^\top \boldsymbol{g}_2)^2}{|\boldsymbol{g}_2^2|} \geq$ $|\boldsymbol{g}_1|^2 - \frac{|\boldsymbol{g}_1|^2 |\boldsymbol{g}_2|^2}{|\boldsymbol{g}_2^2|} = 0$.

However, the PCGrad method can not guarantee a conflict-free update when there are more loss-specific gradients. To illustrate this, let us consider the case of three vectors, $\boldsymbol{g}_1, \boldsymbol{g}_2, \boldsymbol{g}_3$, where each vector conflicts with the other. According to the PCGrad method, we first replace $\boldsymbol{g}_1$ with $\mathcal{O}(\boldsymbol{g}_2, \boldsymbol{g}_1)$, i.e., $\hat{\boldsymbol{g}}_1 = \mathcal{O}(\boldsymbol{g}_2, \boldsymbol{g}_1)$ so that $\hat{\boldsymbol{g}}_1^\top \boldsymbol{g}_2 = 0$. If the updated $\hat{\boldsymbol{g}}_1$ is still conflict with $\boldsymbol{g}_3$, we will then further modify $\hat{\boldsymbol{g}}_1$ as $\hat{\boldsymbol{g}}_1 = \mathcal{O}(\boldsymbol{g}_3, \hat{\boldsymbol{g}}_1)$. However, now we can only guarantee that $\hat{\boldsymbol{g}}_1^\top \boldsymbol{g}_3 = 0$, but we can not guarantee that $\hat{\boldsymbol{g}}_1$ is still not conflicting with $\boldsymbol{g}_2$.

The PCGrad method introduces random selection to mitigate this drawback, as shown in Algorithm 2, but this does not fundamentally solve the problem. Fig. 13 illustrate a simple case for the failure of the PCGrad method where $\boldsymbol{g}_1 = [1.0, 0, 0.1]$, $\boldsymbol{g}_2 = [-0.5, \sqrt{3}/2, 0.1]$, and $\boldsymbol{g}_3 = [-0.5, -\sqrt{3}/2, 0.1]$. The final update vector for the PCGrad method is $\boldsymbol{g}_{\text{PCGrad}} = [-0.351, -0.203, 0.658]$, which conflicts with $\boldsymbol{g}_1$ while the update direction of our ConFIG method is $\boldsymbol{g}_{\text{ConFIG}} = [0, 0, 0.3]$, which is not conflict with any loss-specific gradients.

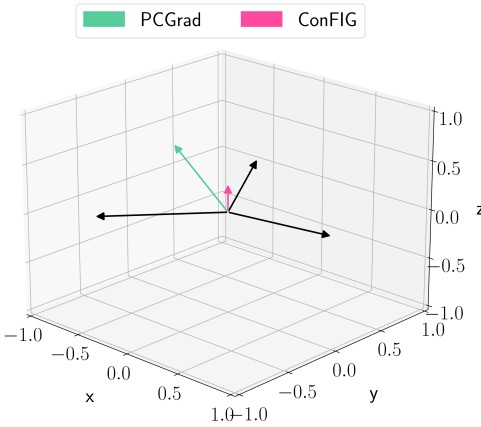

Figure 13: A simple failure model for PCGrad method.

Another feature of the PCGrad method is that the projection length of the final update gradient $\boldsymbol{g}_{\text{PCGrad}}$ on each loss-specific gradient is not equal, i.e., $\boldsymbol{g}_{\text{PCGrad}}^\top \mathcal{U}(\boldsymbol{g}_i) \neq \boldsymbol{g}_{\text{PCGrad}}^\top \mathcal{U}(\boldsymbol{g}_j)$, and $\boldsymbol{g}_{\text{PCGrad}}$ is usually biased to the loss-specific gradient with higher magnitude. We can illustrate this point with two gradient examples:

$$
\begin{aligned}
\boldsymbol{g}_{\text{PCGrad}}^\top \mathcal{U}(\boldsymbol{g}_1) &= (\mathcal{O}(\boldsymbol{g}_1, \boldsymbol{g}_2) + \mathcal{O}(\boldsymbol{g}_2, \boldsymbol{g}_1))^\top \frac{\boldsymbol{g}_1}{|\boldsymbol{g}_1|} \\
&= \mathcal{O}(\boldsymbol{g}_2, \boldsymbol{g}_1)^\top \frac{\boldsymbol{g}_1}{|\boldsymbol{g}_1|} \\
&= \left( \boldsymbol{g}_1 - \frac{\boldsymbol{g}_1^\top \boldsymbol{g}_2}{|\boldsymbol{g}_2|^2} \boldsymbol{g}_2 \right)^\top \frac{\boldsymbol{g}_1}{|\boldsymbol{g}_1|} \\
&= |\boldsymbol{g}_1| - \frac{(\boldsymbol{g}_1^\top \boldsymbol{g}_2)^2}{|\boldsymbol{g}_2|^2 |\boldsymbol{g}_1|} \\
&= |\boldsymbol{g}_1| [1 - \mathcal{S}_c^2(\boldsymbol{g}_1, \boldsymbol{g}_2)].
\end{aligned}
\tag{19}
$$

Similarly,

$$
\boldsymbol{g}_{\text{PCGrad}}^\top \mathcal{U}(\boldsymbol{g}_2) = |\boldsymbol{g}_2| [1 - \mathcal{S}_c^2(\boldsymbol{g}_1, \boldsymbol{g}_2)].
\tag{20}
$$

Thus,

$$
\frac{\boldsymbol{g}_{\text{PCGrad}}^\top \mathcal{U}(\boldsymbol{g}_1)}{\boldsymbol{g}_{\text{PCGrad}}^\top \mathcal{U}(\boldsymbol{g}_2)} = \frac{|\boldsymbol{g}_1|}{|\boldsymbol{g}_2|}
\tag{21}
$$

A similar conclusion also holds for the case with more loss-specific gradients. The essence here is the magnitude of the orthogonal operator which is proportional to the magnitude of loss-specific gradients:

$$
\begin{aligned}
|\mathcal{O}(\boldsymbol{g}_i, \boldsymbol{g}_j)| &= \sqrt{(\boldsymbol{g}_j - \frac{\boldsymbol{g}_i^\top \boldsymbol{g}_j}{|\boldsymbol{g}_i|^2}\boldsymbol{g}_i)^2} \\
&= \sqrt{(|\boldsymbol{g}_j|^2 + \frac{(\boldsymbol{g}_i^\top \boldsymbol{g}_j)^2}{|\boldsymbol{g}_i|^2} - 2\frac{\boldsymbol{g}_i^\top \boldsymbol{g}_j}{|\boldsymbol{g}_i|^2}\boldsymbol{g}_i^\top \boldsymbol{g}_j)} \\
&= \sqrt{|\boldsymbol{g}_j|^2 - \frac{(\boldsymbol{g}_i^\top \boldsymbol{g}_j)^2}{|\boldsymbol{g}_i|^2}} \\
&= \sqrt{|\boldsymbol{g}_j|^2 - |\boldsymbol{g}_j|^2 \mathcal{S}_c^2(\boldsymbol{g}_i, \boldsymbol{g}_j)} \\
&= |\boldsymbol{g}_j|\sqrt{1 - \mathcal{S}_c^2(\boldsymbol{g}_i, \boldsymbol{g}_j)}.
\end{aligned}
\tag{22}
$$

When summing all these orthogonal values together, the final gradient will be biased to the $\mathcal{O}(\boldsymbol{g}_i, \boldsymbol{g}_j)$ with a larger gradient. Meanwhile, since $\mathcal{O}(\boldsymbol{g}_i, \boldsymbol{g}_j)$ is biased towards $\boldsymbol{g}_j$ rather than $\boldsymbol{g}_i$(due to $\mathcal{S}_c(\boldsymbol{g}_i, \mathcal{O}(\boldsymbol{g}_i, \boldsymbol{g}_j)) = 0$), the final gradient will favor to the loss-specific gradient with larger magnitude.

**IMTL-G method.** In contrast to the PCGrad method the IMTL-G method is designed to guarantee an equal projection length, i.e.,

$$
\boldsymbol{g}_{\text{IMTL-G}} \cdot \mathcal{U}(g_i)^\top = \boldsymbol{g}_{\text{IMTL-G}} \cdot \mathcal{U}(g_j)^\top
\tag{23}
$$

To achieve this goal, the IMTL-G method rescales the magnitude of each loss-specific gradient with calculated weights:

$$
\boldsymbol{g}_{\text{IMTL-G}} = \sum_{i=1}^{m} \alpha_i \boldsymbol{g}_i
\tag{24}
$$

$$
\begin{cases}
[\alpha_2, \alpha_3 \cdots \alpha_m] = \boldsymbol{g}_1 \cdot \boldsymbol{U}^\top \cdot (\boldsymbol{D} \cdot \boldsymbol{U}^\top)^{-1} \\
\alpha_1 = 1 - \sum_{i=2}^{m} \alpha_i \\
\boldsymbol{U}^\top = [\mathcal{U}(\boldsymbol{g}_1^\top) - \mathcal{U}(\boldsymbol{g}_2^\top), \mathcal{U}(\boldsymbol{g}_1^\top) - \mathcal{U}(\boldsymbol{g}_3^\top), \cdots, \mathcal{U}(\boldsymbol{g}_1^\top) - \mathcal{U}(\boldsymbol{g}_m^\top)] \\
\boldsymbol{D}^\top = [\boldsymbol{g}_1^\top - \boldsymbol{g}_2^\top, \boldsymbol{g}_1^\top - \boldsymbol{g}_3^\top, \cdots, \boldsymbol{g}_1^\top - \boldsymbol{g}_m^\top]
\end{cases}
\tag{25}
$$

For the two gradient scenario, this will give $[\alpha_1, \alpha_2] = [|\boldsymbol{g}_2|/(|\boldsymbol{g}_1| + |\boldsymbol{g}_2|), |\boldsymbol{g}_1|/(|\boldsymbol{g}_1| + |\boldsymbol{g}_2|)]$, resulting in a same magnitude for two loss-specific gradients as $(|\boldsymbol{g}_1||\boldsymbol{g}_2|)/(|\boldsymbol{g}_1| + |\boldsymbol{g}_2|)$. The final gradient vector doesn't conflict with $\boldsymbol{g}_1$ and $\boldsymbol{g}_2$, as

$$
\begin{aligned}
\boldsymbol{g}_{\text{IMTL-G}}^\top(U)(\boldsymbol{g}_1) &= \frac{|\boldsymbol{g}_1||\boldsymbol{g}_2|}{|\boldsymbol{g}_1| + |\boldsymbol{g}_2|}(\frac{\boldsymbol{g}_1}{|\boldsymbol{g}_1|} + \frac{\boldsymbol{g}_2}{|\boldsymbol{g}_2|})^\top \frac{\boldsymbol{g}_1}{|\boldsymbol{g}_1|} \\
&= \frac{|\boldsymbol{g}_1||\boldsymbol{g}_2|}{|\boldsymbol{g}_1| + |\boldsymbol{g}_2|}(1 + \mathcal{S}_c(\boldsymbol{g}_1, \boldsymbol{g}_2)) \\
&\geq 0
\end{aligned}
\tag{26}
$$

As mentioned in Appendix. 3.2, the direction of the final update gradient for IMTL-G and our ConFIG is the same when there are only two vectors, but their magnitudes differ. For the Adam optimizer, the gradient's absolute magnitude is unimportant since it ultimately rescales the gradient elements. The crucial aspect is how the gradient changes in both direction and magnitude. When the gradient changes rapidly, Adam's corresponding learning rate will decrease, as discussed in Appendix. A.5. Since the direction of the final gradient in IMTL-G and ConFIG is the same, how their magnitudes change ultimately affects their performance with the Adam optimizer. The magnitude of the final gradient with two gradients are $2\sqrt{[1 + \mathcal{S}_c(\boldsymbol{g}_1), \boldsymbol{g}_2)]/2}(|\boldsymbol{g}_1||\boldsymbol{g}_2|)/(|\boldsymbol{g}_1| + |\boldsymbol{g}_2|)$ and $\sqrt{[1 + \mathcal{S}_c(\boldsymbol{g}_1), \boldsymbol{g}_2)]/2}(|\boldsymbol{g}_1| + |\boldsymbol{g}_2|)$ for IMTL-G and ConFIG method, respectively. Thus,

$$
\frac{|\boldsymbol{g}_{\text{IMTL-G}}|}{|\boldsymbol{g}_{\text{ConFIG}}|} = \frac{2(|\boldsymbol{g}_1||\boldsymbol{g}_2|)/(|\boldsymbol{g}_1| + |\boldsymbol{g}_2|)}{|\boldsymbol{g}_1| + |\boldsymbol{g}_2|}
\tag{27}
$$

A significant difference arises when the magnitudes of $\boldsymbol{g}_1$ and $\boldsymbol{g}_2$ are different. For example, if $\boldsymbol{g}_1 >> \boldsymbol{g}_2$, $(|\boldsymbol{g}_1||\boldsymbol{g}_2|)/(|\boldsymbol{g}_1| + |\boldsymbol{g}_2|) \sim \boldsymbol{g}_2$ while $|\boldsymbol{g}_1| + |\boldsymbol{g}_2| \sim \boldsymbol{g}_1$. This means that the magnitude of $\boldsymbol{g}_{\text{IMTL-G}}$ tracks the changes in the magnitude of the smaller vector, whereas our ConFIG method tracks the magnitude of the larger vector. This causes the difference between IMTL-G and our ConFIG methods in two loss-term vector situations.

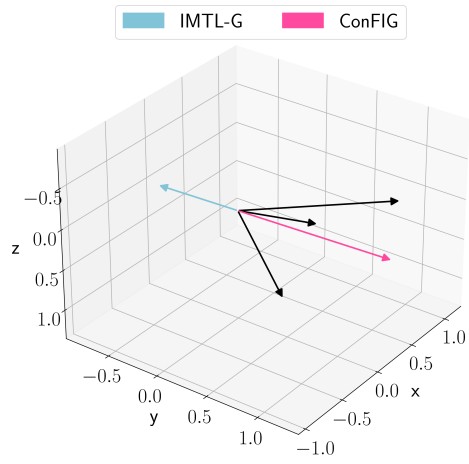

Figure 14: A simple failure model for IMTL-G method.

Another major difference between IMTL-G and our ConFIG methods occurs when more gradient terms are involved. The coefficient $\alpha_i$ of IMTL-G method might have negative values since Eq. 25 only guarantees that $\boldsymbol{g}_{\text{IMTL-G}} \cdot \mathcal{U}(g_i)^\top = \boldsymbol{g}_{\text{IMTL-G}} \cdot \mathcal{U}(g_j)^\top$ but not $\boldsymbol{g}_{\text{IMTL-G}} \cdot \mathcal{U}(g_i)^\top = \boldsymbol{g}_{\text{IMTL-G}} \cdot \mathcal{U}(g_j)^\top \geq 0$. It is easy to find such a situation where the negative coefficient will result in a conflicting update. Fig. 14 illustrates such an example where $\boldsymbol{g}_1 = [0.0412, 0.4295, 0.9394]$, $\boldsymbol{g}_2 = [0.3571, 0.5491, 0.1414]$, and $[0.9823, 0.9361, 0.0552]$. Although both our ConFIG and IMTL-G ensure $\boldsymbol{g}_{\text{IMTL-G}} \cdot \mathcal{U}(g_i)^\top = \boldsymbol{g}_{\text{IMTL-G}} \cdot \mathcal{U}(g_j)^\top$, they are in exactly opposite direction. The obtained update direction for our ConFIG method is $\boldsymbol{g}_{\text{ConFIG}} = [1.5844, 0.4850, 1.4005]$ and the cosine similarity between $\boldsymbol{g}_{\text{ConFIG}}$ and each loss-specific gradient is 0.7086 while IMTL-G method results in a final update gradient of $\boldsymbol{g}_{\text{IMTL-G}} = [-0.7429, -0.2274, -0.6566]$ with a cosine similarity of -0.7086.

## A.3 Existence of Solutions for ConFIG

In Appendix A.2, we illustrate instances where the PCGrad and IMTL-G methods fail. This raises the question of whether ConFIG likewise exhibits failure modes that could impede its performance.

Our ConFIG method is equivalent to solving the following system of linear equations:

$$[\mathcal{U}(\boldsymbol{g}_1), \mathcal{U}(\boldsymbol{g}_2), \cdots, \mathcal{U}(\boldsymbol{g}_m)]^\top \boldsymbol{x} = \mathbf{1}_m, \tag{28}$$

where the solution $\boldsymbol{x}$ is the conflict-free gradient that should be obtained. Our ConFIG method will fail if Eq. 28 has no solution, i.e., if a conflict-free direction in the parameter space does not exist. If we use $\boldsymbol{A}$ and $\boldsymbol{b}$ to denote $[\mathcal{U}(\boldsymbol{g}_1), \mathcal{U}(\boldsymbol{g}_2), \cdots, \mathcal{U}(\boldsymbol{g}_m)]^\top$ and $\mathbf{1}_m$, respectively, Eq. 28 will not have a solution if $R(\boldsymbol{A}) < R[(\boldsymbol{A}|\boldsymbol{b})]$ where $R$ is the rank of a matrix and $(\boldsymbol{A}|\boldsymbol{b})$ is the augmented matrix of $\boldsymbol{A}$ and $\boldsymbol{b}$. Meanwhile, since $R(\boldsymbol{b})=1$, and $R(\boldsymbol{A}) \leq R[(\boldsymbol{A}|\boldsymbol{b})] \leq R(\boldsymbol{A}) + R(\boldsymbol{b})$, i.e., $R(\boldsymbol{A}) \leq R[(\boldsymbol{A}|\boldsymbol{b})] \leq R(\boldsymbol{A}) + 1$, $R[(\boldsymbol{A}|\boldsymbol{b})]$ can only be either $R(\boldsymbol{A})$ or $R(\boldsymbol{A}) + 1$. Thus, Eq. 28 doesn't have solutions when $R[(\boldsymbol{A}|\boldsymbol{b})] = R(\boldsymbol{A}) + 1$.

Assume that each loss-specific gradient has $k$ elements, then the shape of $\boldsymbol{A}$ and $\boldsymbol{b}$ is $m \times k$ and $m \times (k + 1)$, respectively. Thus, we have:

- If $\boldsymbol{A}$ is full rank:
  - If $m > k$: $R(\boldsymbol{A}) = k$;
    * if $(\boldsymbol{A}|\boldsymbol{b})$ is full rank: $R[(\boldsymbol{A}|\boldsymbol{b})] = k + 1$. Eq. 28 doesn't have solutions.

      ∗ if $(\boldsymbol{A}|\boldsymbol{b})$ is not full rank: $R[(\boldsymbol{A}|\boldsymbol{b})] = k$. Eq. 28 has solutions.
    – If $m \leq k$: $R(\boldsymbol{A}) = m$. Since $R[(\boldsymbol{A}|\boldsymbol{b})] \leq min(m, k+1) = m$, thus $R[(\boldsymbol{A}|\boldsymbol{b})] \neq R(\boldsymbol{A}) + 1 = m + 1$. Eq. 28 has solutions.
  • If $\boldsymbol{A}$ is not full rank, i.e., $R(\boldsymbol{A}) = p < min(m,k)$, we will always have $p + 1 \leq min(m, k+1)$. Thus, there is no guarantee that $R[(\boldsymbol{A}|\boldsymbol{b})] \neq m+1$. Thus Eq. 28 doesn't have solutions if $R[(\boldsymbol{A}|\boldsymbol{b})] = p + 1$, and it has solutions if $R[(\boldsymbol{A}|\boldsymbol{b})] = p$.

In practice, the number of parameters in a neural network is always much larger than the number of loss terms, i.e., $k >> m$. In this case, a full rank $\boldsymbol{A}$ means $R(\boldsymbol{A}) = m$, i,e, all the loss-specific gradients are linearly independent. This is true in most situations because if the loss-specific gradients were not linearly independent, we would have $\boldsymbol{g}_j = \sum_{i=1, i \neq j}^{m} \alpha_i \boldsymbol{g}_i$, where $\alpha_i$ is the coefficient. Since $\boldsymbol{g}_i = \partial \mathcal{L}_i / \partial \theta$, this would also imply $\mathcal{L}_j = \sum_{i=1, i \neq j}^{m} \alpha_i \mathcal{L}_i$. However, the relationship between loss functions is usually not linear in practice.

To summarize, during the training procedure, we almost always have $k >> m$ and a full rank $\boldsymbol{A}$, ensuring the existence of a conflict-free direction. It is also worth pointing out that our ConFIG method can still provide a good direction if a conflict-free direction does not exist since the Moore–Penrose pseudoinverse will offer a least squares solution for Eq. 28, making it a "conflict-free" as possible.

## A.4   Proof of the Equivalence

To prove the equivalence between Eq. 2, 3 and Eq. 4, 5 is equivalent to prove that $\mathcal{U}\left[\mathcal{O}(\boldsymbol{g}_1, \boldsymbol{g}_2)\right] + \mathcal{U}\left[\mathcal{O}(\boldsymbol{g}_2, \boldsymbol{g}_1)\right] = k[\mathcal{U}(\boldsymbol{g}_1), \mathcal{U}(\boldsymbol{g}_2)]^{-\top} \mathbf{1}_m$ where k¿0. Thus, we can first calculate

$$[\mathcal{U}(\boldsymbol{g}_1), \mathcal{U}(\boldsymbol{g}_2)]^{\top}[\mathcal{U}[\mathcal{O}(\boldsymbol{g}_1, \boldsymbol{g}_2)] + \mathcal{U}[\mathcal{O}(\boldsymbol{g}_2, \boldsymbol{g}_1)]] = \begin{bmatrix} \mathcal{U}(\boldsymbol{g}_1)^{\top}[\mathcal{U}[\mathcal{O}(\boldsymbol{g}_1, \boldsymbol{g}_2)] + \mathcal{U}[\mathcal{O}(\boldsymbol{g}_2, \boldsymbol{g}_1)]] \\ \mathcal{U}(\boldsymbol{g}_2)^{\top}[\mathcal{U}[\mathcal{O}(\boldsymbol{g}_1, \boldsymbol{g}_2)] + \mathcal{U}[\mathcal{O}(\boldsymbol{g}_2, \boldsymbol{g}_1)]] \end{bmatrix}$$
$$= \begin{bmatrix} \mathcal{U}(\boldsymbol{g}_1)^{\top}\mathcal{U}[\mathcal{O}(\boldsymbol{g}_2, \boldsymbol{g}_1)] \\ \mathcal{U}(\boldsymbol{g}_2)^{\top}\mathcal{U}[\mathcal{O}(\boldsymbol{g}_1, \boldsymbol{g}_2)] \end{bmatrix}. \tag{29}$$

Then, from Eq. 19 we can obtain

$$\mathcal{U}(\boldsymbol{g}_1)^{\top}\mathcal{O}(\boldsymbol{g}_2, \boldsymbol{g}_1) = |\boldsymbol{g}_1|[1 - \mathcal{S}_c^2(\boldsymbol{g}_1, \boldsymbol{g}_2)]. \tag{30}$$

Thus

$$\mathcal{U}(\boldsymbol{g}_1)^{\top}\mathcal{U}[\mathcal{O}(\boldsymbol{g}_2, \boldsymbol{g}_1)] = \frac{|\boldsymbol{g}_1|[1 - \mathcal{S}_c^2(\boldsymbol{g}_1, \boldsymbol{g}_2)]}{|\mathcal{O}(\boldsymbol{g}_2, \boldsymbol{g}_1)|}. \tag{31}$$

From Eq. 22 we can get

$$|\mathcal{O}(\boldsymbol{g}_2, \boldsymbol{g}_1)| = |\boldsymbol{g}_1|\sqrt{1 - \mathcal{S}_c^2(\boldsymbol{g}_1, \boldsymbol{g}_2)}. \tag{32}$$

Put Eq. 32 back to Eq. 31, we get

$$\mathcal{U}(\boldsymbol{g}_1)^{\top}\mathcal{U}[\mathcal{O}(\boldsymbol{g}_2, \boldsymbol{g}_1)] = \sqrt{1 - \mathcal{S}_c^2(\boldsymbol{g}_1, \boldsymbol{g}_2)}. \tag{33}$$

Similarly, we can get

$$\mathcal{U}(\boldsymbol{g}_2)^{\top}\mathcal{U}[\mathcal{O}(\boldsymbol{g}_1, \boldsymbol{g}_2)] = \sqrt{1 - \mathcal{S}_c^2(\boldsymbol{g}_1, \boldsymbol{g}_2)}. \tag{34}$$

Put Eq. 33 and Eq. 34 back to Eq. 29

$$[\mathcal{U}(\boldsymbol{g}_1), \mathcal{U}(\boldsymbol{g}_2)]^{\top}[\mathcal{U}[\mathcal{O}(\boldsymbol{g}_1, \boldsymbol{g}_2)] + \mathcal{U}[\mathcal{O}(\boldsymbol{g}_2, \boldsymbol{g}_1)]] = [\sqrt{1 - \mathcal{S}_c^2(\boldsymbol{g}_1, \boldsymbol{g}_2)}, \sqrt{1 - \mathcal{S}_c^2(\boldsymbol{g}_1, \boldsymbol{g}_2)}]^{\top}, \tag{35}$$

which is

$$\mathcal{U}[\mathcal{O}(\boldsymbol{g}_1, \boldsymbol{g}_2)] + \mathcal{U}[\mathcal{O}(\boldsymbol{g}_2, \boldsymbol{g}_1)] = \sqrt{1 - \mathcal{S}_c^2(\boldsymbol{g}_1, \boldsymbol{g}_2)}[\mathcal{U}(\boldsymbol{g}_1), \mathcal{U}(\boldsymbol{g}_2)]^{-\top}\mathbf{1}_m. \tag{36}$$

## A.5   M-ConFIG and Momentum Strategies

In the current study, we use the Adam optimizer to train baseline PINNs, as shown in Algorithm 3. The green line in the algorithm can be replaced by other methods like PCGrad or the IMTL-G method. The Adam optimizer uses the first momentum $m$ to average the recent update direction and the second momentum $v$ to rescale the learning rate of each parameter. If we want to use the momentum to replace the gradient for the ConFIG update, an intuitive approach could be to calculate the first and second momentum for each loss-specific gradient and use the final rescaled momentum to calculate the input for the ConFIG update, as shown in "MA-ConFIG" in Algorithm 4. However, this approach could result in several issues:

---

**Algorithm 3** ConFIG update with Adam optimizer

---

**Require:** $\theta_0$ (network weights), $\gamma$ (learning rate), $\beta_1$, $\beta_2$, $\epsilon$ (Adam coefficient),
  $\boldsymbol{m}_0 \leftarrow \boldsymbol{0}$ (first momentum), $\boldsymbol{v}_0 \leftarrow \boldsymbol{0}$ (Second momentum),
  All operations on vectors are element-wise except $\mathcal{G}$.
  **for** $t \leftarrow 1$ to $\cdots$ **do**
    $\boldsymbol{g}_c = \mathcal{G}(\nabla_{\theta_{t-1}}\mathcal{L}_1, \nabla_{\theta_{t-1}}\mathcal{L}_2, \cdots, \nabla_{\theta_{t-1}}\mathcal{L}_m)$         ▷ ConFIG update of gradients
    $\boldsymbol{m}_t \leftarrow \beta_1 \boldsymbol{m}_{t-1} + (1 - \beta_1)\boldsymbol{g}_c$         ▷ Update the first momentum
    $\boldsymbol{v}_t \leftarrow \beta_2 \boldsymbol{v}_{t-1} + (1 - \beta_2)\boldsymbol{g}_c^2$         ▷ Update the second momentum
    $\hat{\boldsymbol{m}} \leftarrow \boldsymbol{m}_t/(1 - \beta_1^t)$         ▷ Bias corrections for the first momentum
    $\hat{\boldsymbol{v}} \leftarrow \boldsymbol{v}_t/(1 - \beta_2^t)$         ▷ Bias corrections for the second momentum
    $\theta_i \leftarrow \theta_{t-1} - \gamma\hat{\boldsymbol{m}}/(\sqrt{\hat{\boldsymbol{v}}} + \epsilon)$         ▷ Update weights of the neural network
  **end for**

---

**Algorithm 4** MA-ConFIG

---

**Require:** $\theta_0$ (network weights), $\gamma$ (learning rate), $\beta_1$, $\beta_2$, $\epsilon$ (Adam coefficient),
  $[\boldsymbol{m}_{\boldsymbol{g}_1,0}, \boldsymbol{m}_{\boldsymbol{g}_1,0}, \cdots \boldsymbol{m}_{\boldsymbol{g}_m,0}] \leftarrow [\boldsymbol{0}, \boldsymbol{0}, \cdots, \boldsymbol{0}]$ (first momentum),
  $[\boldsymbol{v}_{\boldsymbol{g}_1,0}, \boldsymbol{v}_{\boldsymbol{g}_1,0}, \cdots \boldsymbol{v}_{\boldsymbol{g}_m,0}] \leftarrow [\boldsymbol{0}, \boldsymbol{0}, \cdots, \boldsymbol{0}]$ (Second momentum), $[t_{\boldsymbol{g}_1}, t_{\boldsymbol{g}_2}, \cdots, t_{\boldsymbol{g}_m}] \leftarrow [0, 0, \cdots, 0]$,
  All operations on vectors are element-wise except $\mathcal{G}$.
  **for** $t \leftarrow 1$ to $\cdots$ **do**
    $i = t\%m + 1$
    $t_{\boldsymbol{g}_i} \leftarrow t_{\boldsymbol{g}_i} + 1$
    $\boldsymbol{g}_i = \nabla_{\theta_{t-1}}\mathcal{L}_i$
    $\boldsymbol{m}_{\boldsymbol{g}_i,t_{\boldsymbol{g}_i}} \leftarrow \beta_1 \boldsymbol{m}_{\boldsymbol{g}_i,t_{\boldsymbol{g}_i}-1} + (1 - \beta_1)\boldsymbol{g}_i$         ▷ Update the first momentum of $\boldsymbol{g}_i$
    $\boldsymbol{v}_{\boldsymbol{g}_i,t_{\boldsymbol{g}_i}} \leftarrow \beta_2 \boldsymbol{v}_{\boldsymbol{g}_i,t_{\boldsymbol{g}_i}-1} + (1 - \beta_2)\boldsymbol{g}_i^2$         ▷ Update the second momentum of $\boldsymbol{g}_i$
    $[\hat{\boldsymbol{m}}_{\boldsymbol{g}_1}, \hat{\boldsymbol{m}}_{\boldsymbol{g}_2}, \cdots, \hat{\boldsymbol{m}}_{\boldsymbol{g}_m}] \leftarrow [\frac{\boldsymbol{m}_{\boldsymbol{g}_1,t_{\boldsymbol{g}_1}}}{1-\beta_1^{t_{\boldsymbol{g}_1}}}, \frac{\boldsymbol{m}_{\boldsymbol{g}_2,t_{\boldsymbol{g}_2}}}{1-\beta_1^{t_{\boldsymbol{g}_2}}}, \cdots, \frac{\boldsymbol{m}_{\boldsymbol{g}_m,t_{\boldsymbol{g}_m}}}{1-\beta_1^{t_{\boldsymbol{g}_m}}}]$   ▷ Bias corrections for first momentum terms
    $[\hat{\boldsymbol{v}}_{\boldsymbol{g}_1}, \hat{\boldsymbol{v}}_{\boldsymbol{g}_2}, \cdots, \hat{\boldsymbol{v}}_{\boldsymbol{g}_m}] \leftarrow [\frac{\boldsymbol{v}_{\boldsymbol{g}_1,t_{\boldsymbol{g}_1}}}{1-\beta_2^{t_{\boldsymbol{g}_1}}}, \frac{\boldsymbol{v}_{\boldsymbol{g}_2,t_{\boldsymbol{g}_2}}}{1-\beta_2^{t_{\boldsymbol{g}_2}}}, \cdots, \frac{\boldsymbol{v}_{\boldsymbol{g}_m,t_{\boldsymbol{g}_m}}}{1-\beta_2^{t_{\boldsymbol{g}_m}}}]$   ▷ Bias corrections for first momentum terms
    $[\hat{\boldsymbol{g}_1}, \hat{\boldsymbol{g}_2}, \cdots, \hat{\boldsymbol{g}_m}] = [\frac{\hat{\boldsymbol{m}}_{\boldsymbol{g}_1}}{\sqrt{\hat{\boldsymbol{v}}_{\boldsymbol{g}_1}}+\varepsilon}, \frac{\hat{\boldsymbol{m}}_{\boldsymbol{g}_2}}{\sqrt{\hat{\boldsymbol{v}}_{\boldsymbol{g}_2}}+\varepsilon}, \cdots, \frac{\hat{\boldsymbol{m}}_{\boldsymbol{g}_m}}{\sqrt{\hat{\boldsymbol{v}}_{\boldsymbol{g}_m}}+\varepsilon}]$         ▷ Rescale the momentum.
    $\theta_i \leftarrow \theta_{t-1} - \gamma\mathcal{G}(\hat{\boldsymbol{g}_1}, \hat{\boldsymbol{g}_2}, \cdots, \hat{\boldsymbol{g}_m})$   ▷ Update weights of the neural network with ConFIG operator
  **end for**

---

- Inappropriate learning rate. Ideally, the Adam method will rescale each element of the update gradient to 1 ($\boldsymbol{m} \sim \boldsymbol{g}$, $\boldsymbol{v} \sim \boldsymbol{g}^2$, $\boldsymbol{m}/\sqrt{\boldsymbol{v}} \sim 1$). If one of the elements of the update gradient repeatedly oscillates between positive and negative, then $m$ becomes smaller while $v$ becomes larger, resulting in a small update step (learning rate) for the corresponding parameter. However, Adam's neat adaptive learning rate adjustment feature will be destroyed if we rescale the momentum before applying the ConFIG operator. This is because ConFIG will alter each parameter's learning rate again by changing the update gradient's direction and magnitude.

- Numerical instability. The current study uses singular value decomposition (SVD) to calculate the pseudoinverse. Our experience shows that rescaling the momentum makes the input

Table 1: Test results of different momentum configurations. "MA-ConFIG" refers to the strategy with the second momentum inside the ConFIG operation, and "NaN" means the training failed due to numerical stability.

|  |  | M-ConFIG | MA-ConFIG |
|---|---|---|---|
| Burgers |  | $(1.277 \pm 0.035) \times 10^{-4}$ | $(1.549 \pm 0.362) \times 10^{-3}$ |
| Schrödinger |  | $(4.292 \pm 1.863) \times 10^{-4}$ | $(2.625 \pm 0.087) \times 10^{-1}$ |
| Kovasznay | $[\mathcal{L}_\mathcal{N}, \mathcal{L}_{\mathcal{BI}}]$ | $(9.777 \pm 0.347) \times 10^{-9}$ | NaN |
| Beltrami |  | $(7.949 \pm 0.384) \times 10^{-5}$ | $(6.658 \pm 0.435) \times 10^{-3}$ |
| Burgers |  | $(1.296 \pm 0.013) \times 10^{-4}$ | $(8.511 \pm 7.835) \times 10^{-3}$ |
| Schrödinger | $[\mathcal{L}_\mathcal{N}, \mathcal{L}_\mathcal{B}, \mathcal{L}_\mathcal{I}]$ | $(1.522 \pm 0.581) \times 10^{-3}$ | NaN |
| Beltrami |  | $(9.103 \pm 1.831) \times 10^{-5}$ | $(5.783 \pm 0.405) \times 10^{-3}$ |

matrix for SVD easily ill-conditioned or has too many repeated singular values, leading to training failure.

Tab. 1 compares the performance of M-ConFIG and MA-ConFIG methods, where M-ConFIG is always stable and significantly better than the MA-ConFIG method.

Note that the M-ConFIG method should not be implemented in combination with Adam, as it is intended to replace Adam's calculation of momentum. Hence, our M-ConFIG implementation uses an SGD optimizer.

## A.6 COMPUTATIONAL COST

To evaluate runtime performance, Fig. 15 compares the relative wall time of all methods w.r.t. the Adam baseline, highlighting the significant advantage of the M-ConFIG on the training cost. The average training cost of M-ConFIG is $0.712\times$ that of the Adam baseline for the two-term case, and $0.557\times$ for the three-term case.

It is also worth noting that the time cost of the ConFIG method is comparable to other gradient-based methods, indicating that the pseudoinverse operation of the ConFIG method does not add significant computational overhead. This study uses the PyTorch implementation, which utilizes singular value decomposition (SVD) to calculate the pseudoinverse. Although the exact computational cost depends on the specific numerical algorithm, the general time complexity of SVD is $\mathcal{O}(nm^2)$ (Grasedyck, 2010) where $n$ is the number of gradient elements (i.e., the number of neural network's parameter) and $m$ is the number of gradients in the current study ($n > m$). This is favorable since the number of network parameters is usually much larger than the number of loss terms, and the time complexity of SVD scales linearly with the former.

On the side of memory, all gradient-based methods, including PCGrad, IMTL-G, and our ConFIG method, usually require more memory compared to the weighting method during training due to the additional cost of storing loss-specific gradients for each loss term in each iteration. The complexity of this additional memory is $\mathcal{O}(nm)$. Meanwhile, although momentum acceleration helps to increase the training speed, it also introduces an increased memory cost with a complexity of $\mathcal{O}(nm)$, as it also needs to store the first and second momentum during the training procedure. This could be a potential challenge when dealing with large-scale problems.

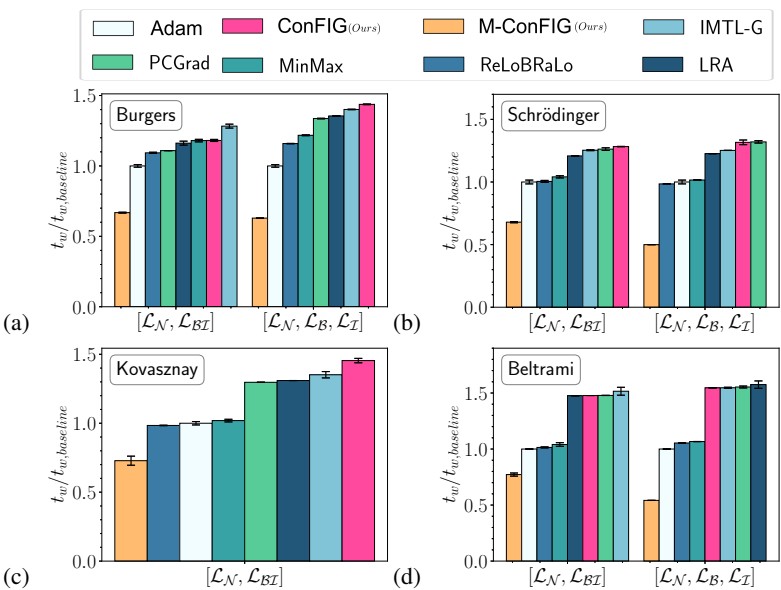

Figure 15: Relative wall time of different methods w.r.t. the Adam baseline for one training iteration.

A.7   PDE DETAILS

**1D Burgers equation.**   Burgers equation is a non-linear and non-trivial model equation describing shock formations. Considering a spatial-temporal field $u(x, t)$, the Burgers equation in one space dimension is

$$\frac{\partial u}{\partial t} + u\frac{\partial u}{\partial x} = \nu\frac{\partial^2 u}{\partial x^2}, \tag{37}$$

where $\nu$ is the viscosity and set as $\nu = 0.01/\pi$ in the current study. The spatial-temporal domain is $t \in [0, 1]$ and $x \in [-1.0, 1.0]$ with the corresponding initial and boundary conditions of

$$\begin{cases} u(x, 0) = -\sin(\pi x) \\ u(+1.0, t) = u(-1.0, t) = 0 \end{cases}. \tag{38}$$

No analytical solution is available for Burgers equation with these given boundary conditions. In the current study, we use numerical solutions from PhiFlow (Holl et al., 2020) as the ground truth solution to evaluate the PINNs' performance.

**1D Schrödinger equation.**   1D Schrödinger equation describes the evaluation of a complex field $h(x, t) = u(x, t) + iv(x, t)$ in one space dimension:

$$i\frac{\partial h}{\partial t} + \frac{1}{2}\frac{\partial^2 h}{\partial x^2} + |h|^2 h = 0, \tag{39}$$

$x \in [-5.0, 5.0]$, $t \in [0, \pi/2]$. It follows a periodic boundary condition and a initial condition of

$$h(0, x) = 2\,\text{sech}(x), \tag{40}$$

Similarly, no analytical solution is available for the 1D Schrödinger equation with given boundary conditions. We use the numerical solution provided by deepXDE (Bajaj et al., 2023) to evaluate the performance of trained PINNs.

**2D Kovasznay flow and 3D Beltrami flow.**   Kovasznay and Beltrami flow are special solutions for Navier-Stokes equations which describe the fluid flow:

$$\frac{\partial \boldsymbol{u}}{\partial t} + (\boldsymbol{u} \cdot \nabla)\boldsymbol{u} = -\nabla p + \frac{1}{Re}\nabla^2 \boldsymbol{u}$$
$$\nabla \cdot \boldsymbol{u} = 0 \tag{41}$$

where $\boldsymbol{u}$ is the velocity vector, $p$ is the pressure and $Re$ is the Reynolds number. Kovasznay flow is a steady case in two space dimensions where the transient term $\partial\boldsymbol{u}/\partial t$ equals 0. The analytical solution of the velocity $\boldsymbol{u}(x, y) = [u_x(x, y), u_y(x, y)]$ and pressure $p(x, y)$ is (Kovasznay, 1948; Truesdell, 1954)

$$\begin{cases} u_x(x, y) & = 1 - e^{\lambda x}\cos(2\pi y), \\ u_y(x, y) & = \frac{\lambda}{2\pi}e^{\lambda x}\sin(2\pi y), \\ p(x, y) & = \frac{1}{2}\left(1 - e^{2\lambda x}\right), \end{cases} \tag{42}$$

where $\lambda = \frac{1}{2\nu} - \sqrt{\frac{1}{4\nu^2} + 4\pi^2}$, $\nu = \frac{1}{Re} = \frac{1}{40}$, $x \in [-0.5, 1]$ and $y \in [-0.5, 1.5]$. Meanwhile, Beltrami flow is an unsteady case in three space dimensions where the velocity $\boldsymbol{u}(x, y, z, t) = [u_x(x, y, z, t), u_y(x, y, z, t), u_z(x, y, z, t)]$ and pressure $p(x, y, z, t)$ follows the analytical solution of (Gromeka, 1881)

$$\begin{cases} u_x(x, y, z, t) = & -a\left[e^{ax}\sin(ay + dz) + e^{az}\cos(ax + dy)\right]e^{-d^2 t} \\ u_y(x, y, z, t) = & -a\left[e^{ay}\sin(az + dx) + e^{ax}\cos(ay + dz)\right]e^{-d^2 t} \\ u_z(x, y, z, t) = & -a\left[e^{az}\sin(ax + dy) + e^{ay}\cos(az + dx)\right]e^{-d^2 t} \\ p(x, y, z, t) = & -\frac{1}{2}a^2\left[e^{2ax} + e^{2ay} + e^{2az}\right. \\ & +2\sin(ax + dy)\cos(az + dx)e^{a(y+z)} \\ & +2\sin(ay + dz)\cos(ax + dy)e^{a(z+x)} \\ & \left.+2\sin(az + dx)\cos(ay + dz)e^{a(x+y)}\right]e^{-2d^2 t} \end{cases}, \tag{43}$$

where $a = d = 1$ when $Re = 1$ in our configuration. The simulation domain size is $x \in [-1, 1]$, $y \in [-1, 1]$, $z \in [-1, 1]$, and $t \in [0, 1]$. The boundary conditions of Kovasznay and Beltrami flow are Dirichlet type, which follows the values of analytical solutions. For Beltrami flow, the initial condition also follows the values of the analytical solutions.

## A.8 PINN-BASED EXAMPLE SOLUTIONS

This section provides examples of the solution domain for each PINN case. Figures 16, 18, 23, and 22 show the mean predictions of the networks compared to the ground truth, the standard deviation, and the squared error distribution for each PINN case. Additionally, Figures 17, 19, and 21 offer a detailed comparison of a sample line in the solution domain for the Burgers equation, Schrödinger equation, and Kovasznay flow case.

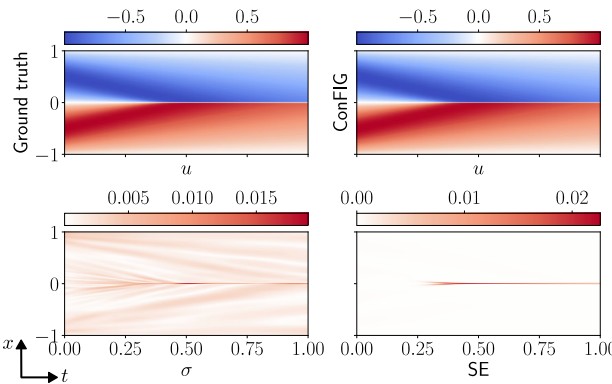

Figure 16: The solution domain of Burgers Equation with corresponding standard deviation($\sigma$) and square error(SE) of the ConFIG method's prediction.

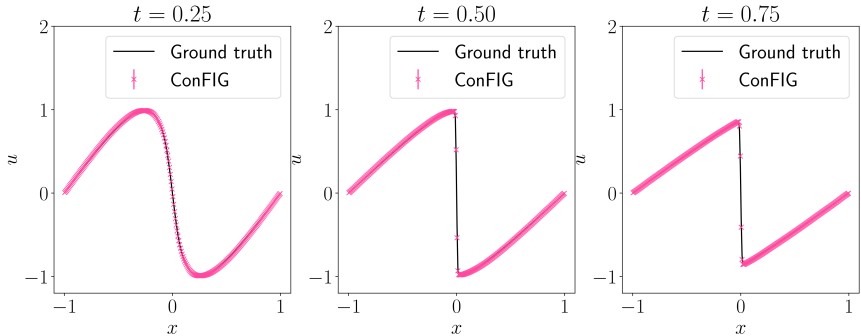

Figure 17: The distribution of $u$ in the solution domain of Burgers equation.

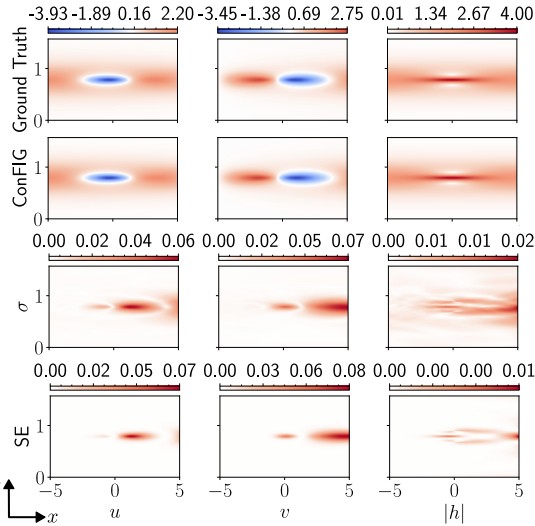

Figure 18: The solution domain of Schrödinger Equation with corresponding standard deviation($\sigma$) and square error(SE) of the ConFIG method's prediction.

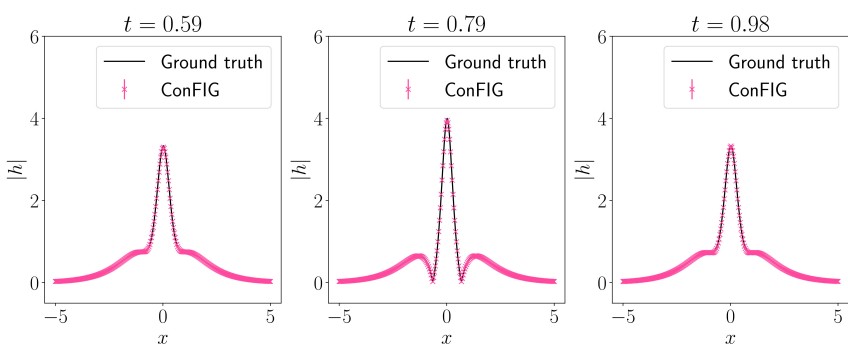

Figure 19: The distribution of $|h|$ in the solution domain of Schrödinger equation.

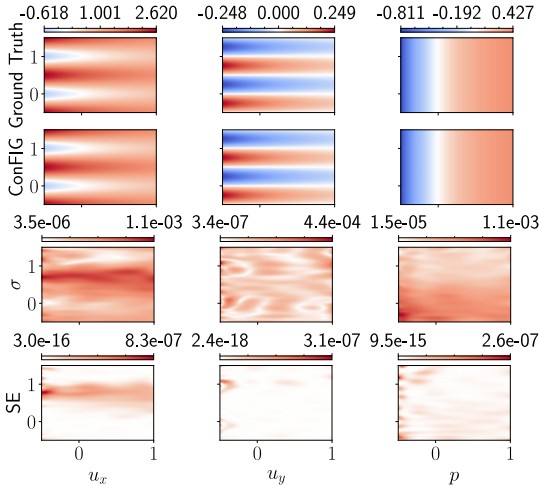

Figure 20: The solution domain of Kovasznay Flow with corresponding standard deviation($\sigma$) and square error(SE) of the ConFIG method's prediction.

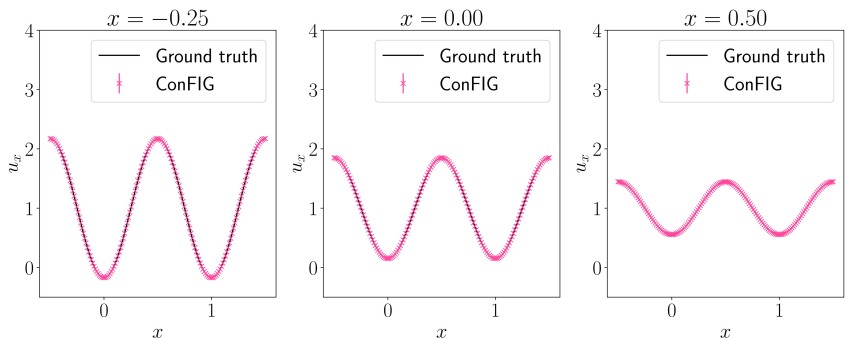

Figure 21: The distribution of $u_x$ in the solution domain of Kovasznay Flow.

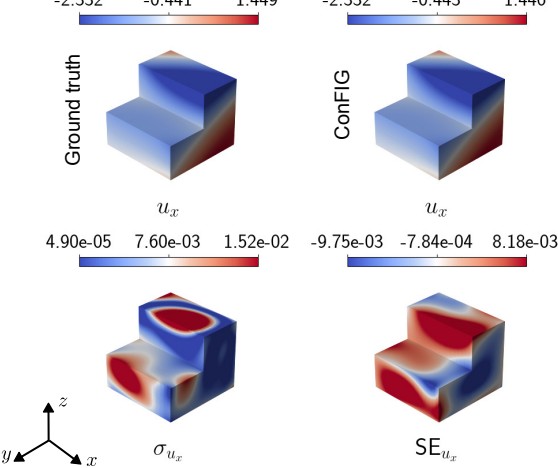

Figure 22: The solution domain of Beltrami Flow when $t = 0.5$ with corresponding standard deviation($\sigma$) and square error(SE) of the ConFIG method's prediction.

## A.9 PINN Evaluations

The exact numerical values and corresponding standard deviations of the PINN experiments are presented in this section. Tables 2 through 5 display the values obtained with the same number of training epochs as depicted in Fig. 4 and Fig. 6. Additionally, Tables 6 through 9 showcase the values obtained with identical wall time, corresponding to the data presented in Fig. 9.

Table 2: Test MSE of PINNs trained for Burgers equation. All values are scaled with $10^{-4}$.

|  | $[\mathcal{L}_{\mathcal{N}}, \mathcal{L}_{\mathcal{BI}}]$ | $[\mathcal{L}_{\mathcal{N}}, \mathcal{L}_{\mathcal{B}}, \mathcal{L}_{\mathcal{I}}]$ |
|---|---|---|
| Adam | $1.484 \pm 0.061$ | |
| PCGrad | $1.344 \pm 0.019$ | $\mathbf{1.279 \pm 0.008}$ |
| IMTL-G | $1.339 \pm 0.024$ | $22.478 \pm 15.008$ |
| MinMax | $1.889 \pm 0.143$ | $1.582 \pm 0.051$ |
| ReLoBRaLo | $1.419 \pm 0.053$ | $1.402 \pm 0.034$ |
| LRA | $353.796 \pm 114.972$ | $444.603 \pm 16.695$ |
| ConFIG | $1.308 \pm 0.008$ | $1.291 \pm 0.039$ |
| M-ConFIG | $\mathbf{1.277 \pm 0.035}$ | $1.296 \pm 0.013$ |

Table 3: Test MSE of PINNs trained for Schrödinger equation. All values are scaled with $10^{-4}$.

|  | $[\mathcal{L}_{\mathcal{N}}, \mathcal{L}_{\mathcal{BI}}]$ | $[\mathcal{L}_{\mathcal{N}}, \mathcal{L}_{\mathcal{B}}, \mathcal{L}_{\mathcal{I}}]$ |
|---|---|---|
| Adam | $3.383 \pm 1.178$ | |
| PCGrad | $1.621 \pm 0.547$ | $\mathbf{0.931 \pm 0.028}$ |
| IMTL-G | $7.891 \pm 3.008$ | $2504.282 \pm 588.560$ |
| MinMax | $45.255 \pm 1.535$ | $45.068 \pm 5.292$ |
| ReLoBRaLo | $3.603 \pm 2.165$ | $2.756 \pm 0.098$ |
| ConFIG | $\mathbf{0.643 \pm 0.227}$ | $1.455 \pm 0.455$ |
| M-ConFIG | $4.292 \pm 1.863$ | $15.217 \pm 5.807$ |

Table 4: Test MSE of PINNs trained for Kovasznay flow. All values are scaled with $10^{-7}$.

|  | $[\mathcal{L}_{\mathcal{N}}, \mathcal{L}_{\mathcal{B}}]$ |
|---|---|
| Adam | $1.044 \pm 0.405$ |
| PCGrad | $0.799 \pm 0.083$ |
| IMTL-G | $\mathbf{0.096 \pm 0.012}$ |
| MinMax | $7.743 \pm 1.197$ |
| ReLoBRaLo | $1.148 \pm 0.359$ |
| LRA | $3.595 \pm 3.582$ |
| ConFIG | $0.126 \pm 0.048$ |
| M-ConFIG | $0.098 \pm 0.003$ |

Table 5: Test MSE of PINNs trained for Beltrami flow. All values are scaled with $10^{-4}$.

|  | $[\mathcal{L}_\mathcal{N}, \mathcal{L}_{\mathcal{BI}}]$ | $[\mathcal{L}_\mathcal{N}, \mathcal{L}_\mathcal{B}, \mathcal{L}_\mathcal{I}]$ |
| --- | --- | --- |
| Adam | $1.112 \pm 0.119$ | |
| PCGrad | $1.037 \pm 0.119$ | $0.757 \pm 0.082$ |
| IMTL-G | $0.568 \pm 0.078$ | $0.932 \pm 0.228$ |
| MinMax | $1.851 \pm 0.100$ | $2.215 \pm 0.192$ |
| ReLoBRaLo | $1.197 \pm 0.029$ | $1.028 \pm 0.107$ |
| LRA | $0.708 \pm 0.047$ | $0.843 \pm 0.146$ |
| ConFIG | $\mathbf{0.617 \pm 0.112}$ | $\mathbf{0.571 \pm 0.090}$ |
| M-ConFIG | $0.795 \pm 0.038$ | $0.910 \pm 0.183$ |

Table 6: Test MSE of PINNs trained for Burgers equation. All values are collected after the same wall time and scaled with $10^{-4}$.

|  | $[\mathcal{L}_\mathcal{N}, \mathcal{L}_{\mathcal{BI}}]$ | $[\mathcal{L}_\mathcal{N}, \mathcal{L}_\mathcal{B}, \mathcal{L}_\mathcal{I}]$ |
| --- | --- | --- |
| Adam | $1.515 \pm 0.079$ | $1.546 \pm 0.100$ |
| PCGrad | $1.476 \pm 0.022$ | $1.300 \pm 0.023$ |
| IMTL-G | $1.356 \pm 0.046$ | $82.446 \pm 76.016$ |
| MinMax | $3.058 \pm 0.379$ | $2.030 \pm 0.153$ |
| ReLoBRaLo | $1.449 \pm 0.033$ | $1.492 \pm 0.099$ |
| LRA | $376.449 \pm 100.599$ | $470.108 \pm 28.172$ |
| ConFIG | $1.330 \pm 0.015$ | $1.665 \pm 0.186$ |
| M-ConFIG | $\mathbf{1.277 \pm 0.035}$ | $\mathbf{1.296 \pm 0.013}$ |

Table 7: Test MSE of PINNs trained for Schrödinger equation. All values are collected after the same wall time and scaled with $10^{-3}$.

|  | $[\mathcal{L}_\mathcal{N}, \mathcal{L}_{\mathcal{BI}}]$ | $[\mathcal{L}_\mathcal{N}, \mathcal{L}_\mathcal{B}, \mathcal{L}_\mathcal{I}]$ |
| --- | --- | --- |
| Adam | $2.169 \pm 0.584$ | $5.832 \pm 1.421$ |
| PCGrad | $1.505 \pm 0.559$ | $2.271 \pm 0.205$ |
| IMTL-G | $8.844 \pm 4.263$ | $291.037 \pm 3.682$ |
| MinMax | $28.195 \pm 1.616$ | $49.635 \pm 2.888$ |
| ReLoBRaLo | $2.369 \pm 0.829$ | $15.230 \pm 7.691$ |
| LRA | $316.549 \pm 8.696$ | $315.606 \pm 8.969$ |
| ConFIG | $0.586 \pm 0.256$ | $4.636 \pm 0.932$ |
| M-ConFIG | $\mathbf{0.429 \pm 0.186}$ | $\mathbf{1.522 \pm 0.581}$ |

Table 8: Test MSE of PINNs trained for Kovasznay flow. All values are scaled with $10^{-7}$.

|  | $[\mathcal{L}_\mathcal{N}, \mathcal{L}_\mathcal{B}]$ |
| --- | --- |
| Adam | $4.037 \pm 1.078$ |
| PCGrad | $2.723 \pm 0.421$ |
| IMTL-G | $0.419 \pm 0.058$ |
| MinMax | $261.628 \pm 43.303$ |
| ReLoBRaLo | $3.291 \pm 0.847$ |
| LRA | $13.554 \pm 13.773$ |
| ConFIG | $0.631 \pm 0.194$ |
| M-ConFIG | $\mathbf{0.098 \pm 0.003}$ |

Table 9: Test MSE of PINNs trained for Beltrami flow. All values are collected after the same wall time and scaled with $10^{-4}$.

|  | $[\mathcal{L_N}, \mathcal{L_{BI}}]$ | $[\mathcal{L_N}, \mathcal{L_B}, \mathcal{L_I}]$ |
|---|---|---|
| Adam | $2.318 \pm 0.270$ | $3.648 \pm 0.244$ |
| PCGrad | $2.206 \pm 0.180$ | $2.351 \pm 0.212$ |
| IMTL-G | $1.054 \pm 0.156$ | $2.194 \pm 0.465$ |
| MinMax | $5.422 \pm 0.300$ | $15.754 \pm 1.546$ |
| ReLoBRaLo | $2.605 \pm 0.056$ | $2.769 \pm 0.202$ |
| LRA | $1.194 \pm 0.139$ | $2.259 \pm 0.812$ |
| ConFIG | $1.130 \pm 0.138$ | $1.451 \pm 0.181$ |
| M-ConFIG | $\textbf{0.795} \pm \textbf{0.038}$ | $\textbf{0.910} \pm \textbf{0.183}$ |

Table 10: Test MSE of PINNs trained for Burgers equation with the ConFIG method using different direction weights. All values are scaled with $10^{-4}$.

|  | $[\mathcal{L_N}, \mathcal{L_{BI}}]$ | $[\mathcal{L_N}, \mathcal{L_B}, \mathcal{L_I}]$ |
|---|---|---|
| MinMax | $2.444 \pm 0.312$ | $45.068 \pm 5.292$ |
| ReLoBRaLo | $1.310 \pm 0.007$ | $1.315 \pm 0.048$ |
| LRA | $449.658 \pm 42.392$ | $657.258 \pm 45.487$ |
| Equal | $\textbf{1.308} \pm \textbf{0.008}$ | $\textbf{1.291} \pm \textbf{0.039}$ |

Table 11: Test MSE of PINNs trained for Schrodinger equation with the ConFIG method using different direction weights. All values are scaled with $10^{-4}$.

|  | $[\mathcal{L_N}, \mathcal{L_{BI}}]$ | $[\mathcal{L_N}, \mathcal{L_B}, \mathcal{L_I}]$ |
|---|---|---|
| MinMax | $2.444 \pm 0.312$ | $3.733 \pm 1.351$ |
| ReLoBRaLo | $1.048 \pm 0.404$ | $2.829 \pm 1.444$ |
| LRA | $25.417 \pm 6.768$ | $3118.378 \pm 45.723$ |
| Equal | $\textbf{6.429} \pm \textbf{2.269}$ | $\textbf{1.455} \pm \textbf{0.455}$ |

Table 12: Test MSE of PINNs trained for Kovasznay flow with the ConFIG method using different direction weights. All values are scaled with $10^{-7}$.

|  | $[\mathcal{L_N}, \mathcal{L_{BI}}]$ |
|---|---|
| MinMax | $7.487 \pm 2.545$ |
| ReLoBRaLo | $\textbf{0.104} \pm \textbf{0.008}$ |
| LRA | $(9.047 \pm 2.052) \times 10^5$ |
| Equal | $0.126 \pm 0.048$ |

Table 13: Test MSE of PINNs trained for Beltrami flow with the ConFIG method using different direction weights. All values are scaled with $10^{-4}$.

|  | $[\mathcal{L_N}, \mathcal{L_{BI}}]$ | $[\mathcal{L_N}, \mathcal{L_B}, \mathcal{L_I}]$ |
|---|---|---|
| MinMax | $1.903 \pm 0.271$ | $1.597 \pm 0.039$ |
| ReLoBRaLo | $0.658 \pm 0.157$ | $0.575 \pm 0.059$ |
| LRA | $155.617 \pm 55.853$ | $15.754 \pm 5.049$ |
| Equal | $\textbf{0.617} \pm \textbf{0.112}$ | $\textbf{0.571} \pm \textbf{0.090}$ |

A.10 RESULTS OF CELEBA MTL EXPERIMENTS

Table 14- 19 presents the numerical values and corresponding standard deviations for the CelebA MTL experiments, aligning with the results in Fig. 11 and Fig. 12. The best performance is determined by selecting the best average performance across different tasks during training, which is also the value shown in Fig. 11. The average performance is calculated based on the results from the last 5 epochs, during which most methods have converged. For both best and average performance, our ConFIG method surpasses all other methods in $MR$ metrics. For the performance in $\overline{F_1}$, it tied for first with the FAMO method in best performance and ranked second in the average performance.

Table 14: $MR$ performance of different methods in CelebA MTL experiments

|  | Best performance | Average performance |
|---|---|---|
| PCGrad | $6.425 \pm 0.595$ | $5.475 \pm 0.575$ |
| IMTL-G | $6.058 \pm 0.671$ | $5.475 \pm 0.125$ |
| FAMO | $6.233 \pm 0.085$ | $4.875 \pm 0.100$ |
| NASHMTL | $6.158 \pm 0.242$ | $5.612 \pm 1.237$ |
| RLW | $6.300 \pm 0.602$ | $4.850 \pm 1.050$ |
| CAGrad | $6.767 \pm 0.274$ | $7.387 \pm 1.062$ |
| GRADDROP | $6.733 \pm 0.309$ | $7.375 \pm 0.000$ |
| DWA | $6.908 \pm 0.450$ | $8.500 \pm 0.000$ |
| LS | $7.383 \pm 0.112$ | $8.675 \pm 0.075$ |
| UW | $7.692 \pm 0.051$ | $8.238 \pm 0.438$ |
| M-ConFIG 30 | $5.833 \pm 0.342$ | $7.175 \pm 0.525$ |
| ConFIG | $\mathbf{5.508 \pm 0.392}$ | $\mathbf{4.362 \pm 0.812}$ |

Table 15: $\overline{F_1}$ performance of different methods in CelebA MTL experiments

|  | Best performance | Average performance |
|---|---|---|
| PCGrad | $0.681 \pm 0.003$ | $0.665 \pm 0.007$ |
| IMTL-G | $0.680 \pm 0.003$ | $0.660 \pm 0.003$ |
| FAMO | $0.686 \pm 0.004$ | $\mathbf{0.673 \pm 0.002}$ |
| NASHMTL | $0.683 \pm 0.006$ | $0.642 \pm 0.026$ |
| RLW | $0.680 \pm 0.002$ | $0.663 \pm 0.006$ |
| CAGrad | $0.672 \pm 0.010$ | $0.649 \pm 0.008$ |
| GRADDROP | $0.661 \pm 0.005$ | $0.644 \pm 0.010$ |
| DWA | $0.672 \pm 0.003$ | $0.644 \pm 0.004$ |
| LS | $0.676 \pm 0.004$ | $0.639 \pm 0.004$ |
| UW | $0.675 \pm 0.009$ | $0.641 \pm 0.006$ |
| M-ConFIG 30 | $\mathbf{0.694 \pm 0.005}$ | $0.655 \pm 0.006$ |
| ConFIG | $0.686 \pm 0.006$ | $0.671 \pm 0.003$ |

Table 16: The $\overline{F_1}$ performance of the M-ConFIG and ConFIG method with different numbers of tasks in the CelebA experiment.

| $n_{updates}$ | Best performance | | Average performance | |
|---|---|---|---|---|
|  | ConFIG | M-ConFIG | ConFIG | M-ConFIG |
| 5 | $0.697 \pm 0.007$ | $0.664 \pm 0.005$ | $0.645 \pm 0.006$ | $0.638 \pm 0.003$ |
| 10 | $0.695 \pm 0.006$ | $0.536 \pm 0.057$ | $0.668 \pm 0.013$ | $0.488 \pm 0.055$ |
| 20 | $0.684 \pm 0.008$ | $0.455 \pm 0.032$ | $0.660 \pm 0.007$ | $0.408 \pm 0.015$ |
| 30 | $0.667 \pm 0.006$ | $0.429 \pm 0.023$ | $0.648 \pm 0.006$ | $0.383 \pm 0.025$ |
| 40 | $0.686 \pm 0.006$ | $0.423 \pm 0.026$ | $0.671 \pm 0.003$ | $0.383 \pm 0.037$ |

Table 17: The training time of the M-ConFIG and ConFIG method with different numbers of tasks in the CelebA experiment.

| $n_{tasks}$ | ConFIG | M-ConFIG |
|---|---|---|
| 5 | $180.333 \pm 0.471$ | $67.436 \pm 0.267$ |
| 10 | $288.000 \pm 0.816$ | $73.588 \pm 1.262$ |
| 20 | $561.000 \pm 0.816$ | $84.252 \pm 0.919$ |
| 30 | $861.000 \pm 0.816$ | $110.454 \pm 0.487$ |
| 40 | $1179.000 \pm 2.449$ | $167.423 \pm 0.427$ |

Table 18: The $\overline{F}_1$ performance METof the M-ConFIG and ConFIG method with different numbers of gradient updates in the CelebA experiment (40 tasks).

| $n_{updates}$ | Best performance | | Average performance | |
|---|---|---|---|---|
| | ConFIG | M-ConFIG | ConFIG | M-ConFIG |
| 1 | | $0.423 \pm 0.026$ | | $0.383 \pm 0.037$ |
| 5 | | $0.458 \pm 0.006$ | | $0.406 \pm 0.008$ |
| 10 | | $0.570 \pm 0.049$ | | $0.517 \pm 0.049$ |
| 20 | $0.686 \pm 0.006$ | $0.681 \pm 0.006$ | $0.671 \pm 0.003$ | $0.668 \pm 0.008$ |
| 30 | | $0.694 \pm 0.005$ | | $0.678 \pm 0.006$ |
| 40 | | $0.682 \pm 0.007$ | | $0.654 \pm 0.015$ |

Table 19: The training time of the M-ConFIG and ConFIG method with different numbers of gradient updates in the CelebA experiment (40 tasks).

| $n_{updates}$ | ConFIG | M-ConFIG |
|---|---|---|
| 1 | | $310.000 \pm 0.000$ |
| 5 | | $492.667 \pm 1.247$ |
| 10 | | $723.000 \pm 0.816$ |
| 20 | $2090.618 \pm 5.627$ | $1185.667 \pm 6.018$ |
| 30 | | $1647.667 \pm 7.587$ |
| 40 | | $2084.667 \pm 24.635$ |

### A.11 ADDITIONAL RESULTS ON CHALLENGING PDES

To evaluate the performance of our method for challenging and high-dimensional test cases, we apply it to three problems from a recent benchmark for PINNs (Hao et al., 2023). In the following, we perform a comprehensive investigation of the performance of our method in solving these problems.

#### A.11.1 HIGH DIMENSIONAL PDES

We introduce the N-dimensional Poisson equation as the test case for high-dimensional problems. The governing PDE is

$$-\nabla^2 u = \frac{\pi^2}{4} \sum_{i=1}^{n} \sin(\frac{\pi}{2} x_i), \tag{44}$$

where $n$ is the number of dimensionality. In the current experiment, we choose $n = 5$ and set the spatial domain as $x \in [0,1]^5$ following the configuration of Hao et al. (2023). The ground truth solution is

$$u = \sum_{i=1}^{n} \sin(\frac{\pi}{2} x_i). \tag{45}$$

The PNd problem employs Dirichlet boundary conditions with the boundary value equal to the analytical solution.

Tab. 20 and Fig. 23 summarize the test results of different methods. Our M-ConFIG method ranks first among all methods, followed by the ConFIG methods, showing the superiority of our methods in dealing with high-dimensional problems.

Table 20: Test MSE of PINNs trained for PNd equation. All values $\times 10^{-6}$.

|  | $[\mathcal{L}_\mathcal{N}, \mathcal{L}_\mathcal{B}]$ |
|---|---|
| Adam | $1.916 \pm 0.284$ |
| PCGrad | $1.313 \pm 0.097$ |
| IMTL-G | $0.520 \pm 0.123$ |
| MinMax | $4.604 \pm 0.271$ |
| ReLoBRaLo | $2.265 \pm 0.263$ |
| LRA | $0.639 \pm 0.340$ |
| ConFIG | $0.461 \pm 0.141$ |
| M-ConFIG | $\mathbf{0.415 \pm 0.052}$ |

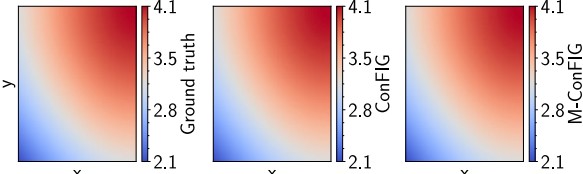

Figure 23: The solution domain of PNd problem on $[x, y, 0.5, 0.5, 0.5]$ plane.

#### A.11.2 MULTI-SCALE PROBLEMS

We choose the multi-scale heat transfer (HeatMS) problem as a strongly anisotropic test case. We set different heat-transfer coefficients in different spatial directions to give the solution different scales in each direction. Following the configuration of Hao et al. (2023), the governing equation is

$$\frac{\partial u}{\partial t} - \frac{1}{(500\pi)^2} \frac{\partial^2 u}{\partial x^2} - \frac{1}{\pi^2} \frac{\partial^2 u}{\partial y^2} = 0, \tag{46}$$

with the initial condition of $u(x, y, 0) = \sin(20\pi x)\sin(\pi y)$ and boundary condition of $u(x, y, t) = 0$. The spatial time domain is $x \in [0, 1]$, $y \in [0, 1]$ and $t \in [0, 5]$. Fig. 24 illustrates a sample solution of the ground truth when $t = 3.0s$.

Tab. 21 and Fig. 25 show the predictions by different methods. While all of the methods struggle to fully resolve the anisotropic solutions, as shown in Fig. 25, our M-ConFIG method still ranks first. This test case illustrates that the scaling issues of anisotropic PDEs can not be solved purely by finding a better balance between different loss terms. Nonetheless, ConFIG fares on-par with other methods, and the momentum terms of M-ConFIG help to partially address the scaling of the HeatMS test case.

Table 21: Test MSE of PINNs trained for HeatMS problem. All values are scaled with $10^{-3}$.

|  | $[\mathcal{L}_\mathcal{N}, \mathcal{L}_{\mathcal{BI}}]$ | $[\mathcal{L}_\mathcal{N}, \mathcal{L}_\mathcal{B}, \mathcal{L}_\mathcal{I}]$ |
|---|---|---|
| Adam | $7.612 \pm 0.004$ | |
| PCGrad | $7.600 \pm 0.012$ | $7.603 \pm 0.012$ |
| IMTL-G | $6.693 \pm 0.241$ | $38.529 \pm 43.666$ |
| MinMax | $7.595 \pm 0.055$ | $7.243 \pm 0.458$ |
| ReLoBRaLo | $7.585 \pm 0.039$ | $7.558 \pm 0.041$ |
| LRA | $7.654 \pm 0.004$ | $7.652 \pm 0.000$ |
| ConFIG | $7.529 \pm 0.074$ | $7.585 \pm 0.020$ |
| M-ConFIG | $\mathbf{5.978 \pm 0.092}$ | $\mathbf{7.147 \pm 0.001}$ |

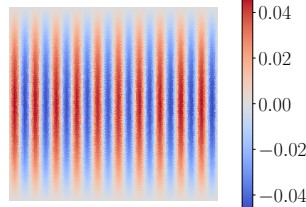

Figure 24: The ground truth solution of HeatMS problem when $t = 3.0s$.

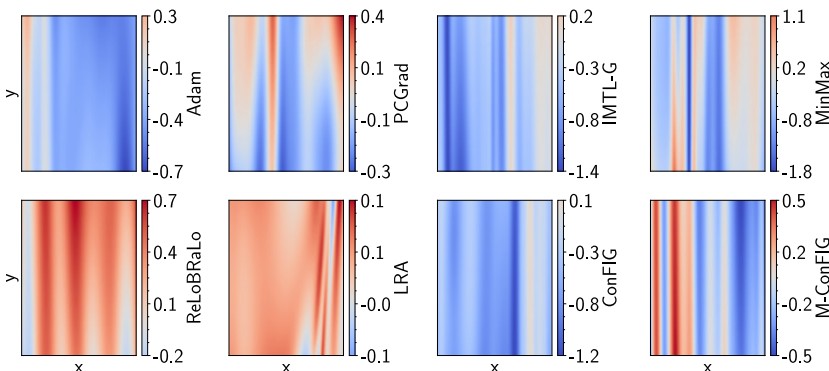

Figure 25: The predictions of PINNs for HeatMS problem when $t = 3.0s$ (two-loss scenario, values scaled with $10^{-2}$).

### A.11.3 CHAOTIC PROBLEMS

To test the potential limitations of the proposed methods, we target the KS equation as a representative of chaotic problems. Starting from an analytic initial state, the solution of the KS equation will gradually develop into a chaotic state due to the strong nonlinearity, dissipative, and destabilizing effects. Following the configuration of Hao et al. (2023), the governing equation is

$$\frac{\partial u}{\partial t} + \frac{100}{16} u \frac{\partial u}{\partial x} + \frac{100}{16^2} \frac{\partial^2 u}{\partial x^2} + \frac{100}{16^4} \frac{\partial^4 u}{\partial x^4} = 0, \tag{47}$$

with an initial condition of $u(x, 0) = \cos(x)(1 + \sin(x))$ and periodic boundary conditions. The spatial time domain is $x \in [0, 2\pi]$ and $t \in [0, 1]$.

Table 22: Test MSE of PINNs for the KS equation.

| | $[\mathcal{L}_{\mathcal{N}}, \mathcal{L}_{\mathcal{BI}}]$ | $[\mathcal{L}_{\mathcal{N}}, \mathcal{L}_{\mathcal{B}}, \mathcal{L}_{\mathcal{I}}]$ |
|---|---|---|
| Adam | $1.094 \pm 0.002$ | $1.094 \pm 0.002$ |
| PCGrad | $1.089 \pm 0.008$ | $1.061 \pm 0.002$ |
| IMTL-G | $1.110 \pm 0.016$ | $1.089 \pm 0.031$ |
| MinMax | $1.072 \pm 0.000$ | $1.084 \pm 0.007$ |
| ReLoBRaLo | $1.074 \pm 0.019$ | $1.080 \pm 0.016$ |
| LRA | $1.099 \pm 0.039$ | $1.098 \pm 0.027$ |
| ConFIG | $1.089 \pm 0.004$ | $1.062 \pm 0.007$ |
| M-ConFIG | $1.123 \pm 0.011$ | $1.118 \pm 0.008$ |

Tab. 22 shows the predictions by different methods. The evaluation shows that no methods succeeds to capture the transition of chaos and generate smooth solutions. This agrees with the conclusions in Hao et al. (2023)'s, where likewise no method under consideration manages to converge to an accurate solution.

The inherent challenge of solving the KS equations is a combination of several factors. On the one hand, the negative diffusion term $\partial^2 u/\partial x^2$ injects energy into the system, inducing instability and complex structures. On the other hand, The dissipative term $\partial^4 u/\partial x^4$ removes energy at small scales, formulating energy cascade across different scales. This energy cascade causes huge difficulties for numerical methods since the fine-scale structures must be resolved accurately. From the PINN side, this requires the network to have the ability to estimate the derivatives accurately on different scales via auto differentiation. The balance between different loss terms, as all the benchmark methods are trying to achieve, is not helpful in addressing this fundamental issue. Hence, this test case serves as a failure case, highlighting that improved optimizers alone do not suffice to address these challenges. It will be an interesting topic of future work to evaluate their effectiveness in combination with other changes, such as improvements on the architecture side.

### A.12 TRAINING DETAILS

**PINNs.** The training of PINNs in the current research follows the established conventions. The neural networks are fully connected with 4 hidden layers and 50 channels per layer. The activation function is the `tanh` function, and all weights are initialized with Xavier initialization (Glorot & Bengio, 2010). Data points are sampled using Latin-hypercube sampling and updated in each iteration. The extended training run case shown in Fig. 10 uses a constant learning rate of $10^{-4}$. All other cases follow a cosine decay strategy with the initial and final learning rate of $10^{-3}$ and $10^{-4}$, respectively. We also add a learning rate warm-up of 100 epochs for each training. All the methods except M-ConFIG use the Adam optimizer. The hyper-parameters of the Adam optimizer are set as $\beta_1=0.9$, $\beta_2 = 0.999$, and $\varepsilon = 10^{-8}$, respectively. The number of data points and training epochs for each case are listed in Tab. 23.

**MTL.** Our experiments are based on the official test code of the FAMO method, and our ConFIG method implemented in the corresponding framework. For the details of the configurations, please refer to the original FAMO paper (Liu et al., 2023) and its official repository: `https://github.com/Cranial-XIX/FAMO` (MIT License).

**Compute resources.** All the experiments in this study were conducted using an NVIDIA RTX A5000 GPU with 24 GB of memory. Each PINN experiment completes training within a few hours on a typical GPU with more than 4 GB of memory. For the CelebA MTL test, a GPU with more than 12 GB of memory is required, and a single training run takes ca. 1-2 days.

Table 23: The number of data points and training epochs for PINNs' experiments.

| | Number of data points | | | Epochs |
|---|---|---|---|---|
| | $n_{\mathcal{N}}$ | $n_{\mathcal{B}}$ | $n_{\mathcal{I}}$ | |
| Burgers equation | 10000 | 250 | 250 | $3 \times 10^4$ |
| Schrödinger equation | 20000 | 500 | 500 | $10^5$ |
| Kovasznay flow | 20000 | 1000 | | $10^5$ |
| Beltrami flow | 25000 | 5000 | 5000 | $10^5$ |
| PNd | 20000 | 5000 | | $10^5$ |
| HeatMS | 20000 | 2000 | 2000 | $10^5$ |
| KS | 20000 | 500 | 500 | $10^5$ |

### A.13 ABLATION STUDY ON TRAINING HYPERPARAMETER

Although we utilize a momentum-based optimizer for the training, the learning rate may still affect the training process, especially considering that our ConFIG and M-ConFIG methods consistently change the gradient vector during the training. Thus, we perform an ablation study on different learning rates. As shown in Tab. 24 and Tab. 25, our methods consistently outperform other methods with different learning rate configurations.

Table 24: The performance of different methods with different cosine decay learning rates in Burgers two-loss test

| | $10^{-3} \rightarrow 10^{-4}$ | $10^{-4} \rightarrow 10^{-5}$ |
|---|---|---|
| Adam | $1.484 \pm 0.061$ | $3.076 \pm 1.201$ |
| PCGrad | $1.344 \pm 0.019$ | $2.223 \pm 0.355$ |
| IMTL-G | $1.339 \pm 0.024$ | $1.688 \pm 0.018$ |
| MinMax | $1.889 \pm 0.143$ | $3.980 \pm 0.798$ |
| ReLoBRaLo | $1.419 \pm 0.053$ | $5.057 \pm 1.821$ |
| LRA | $353.796 \pm 114.972$ | $819.730 \pm 56.925$ |
| ConFIG | $1.308 \pm 0.008$ | $1.887 \pm 0.283$ |
| M-ConFIG | $\textbf{1.277} \pm \textbf{0.035}$ | $\textbf{1.681} \pm \textbf{0.106}$ |

Table 25: The performance of different methods with different constant learning rates in Burgers two-loss test

| | $\gamma = 10^{-3}$ | $\gamma = 10^{-4}$ | $\gamma = 10^{-5}$ |
|---|---|---|---|
| Adam | $1.398 \pm 0.021$ | $3.076 \pm 1.201$ | $420.511 \pm 95.578$ |
| PCGrad | $1.398 \pm 0.018$ | $2.223 \pm 0.355$ | $51.664 \pm 7.753$ |
| IMTL-G | $1.385 \pm 0.042$ | $1.688 \pm 0.018$ | $192.151 \pm 123.903$ |
| MinMax | $1.889 \pm 0.120$ | $3.980 \pm 0.798$ | $7.078 \pm 1.672$ |
| ReLoBRaLo | $1.477 \pm 0.061$ | $5.057 \pm 1.821$ | $10.333 \pm 1.905$ |
| LRA | $367.944 \pm 98.322$ | $819.730 \pm 56.925$ | $895.965 \pm 1.671$ |
| ConFIG | $1.373 \pm 0.006$ | $1.887 \pm 0.283$ | $103.239 \pm 40.287$ |
| M-ConFIG | $\textbf{1.354} \pm \textbf{0.019}$ | $\textbf{1.681} \pm \textbf{0.106}$ | $\textbf{4.097} \pm \textbf{1.310}$ |

In the current study, the points are sampled dynamically from the internal domain and boundaries during training. Thus, the number of data samples represents the training batch size. Meanwhile, each loss-specific gradient is evaluated through the data samples at the internal domain and boundaries. Thus, the number of sample points and the relative ratio between the number of points at the boundary and the internal domain may affect the quality of the gradient, further affecting the final training results. Here, we also perform an ablation study on the data samples, and the results are

summarized in Fig. 26, Tab. 26 and Tab. 27. Our methods consistently outperform other methods with different configurations of data samples, showing the robustness of our methods.

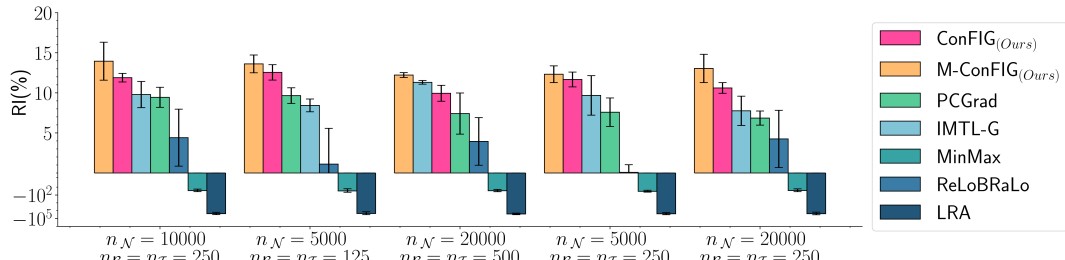

Figure 26: The relative improvements of different methods with different numbers and different ratios of data samples during training

Table 26: The performance of different methods with different numbers of data samples during training

|  | $n_{\mathcal{N}} = 5000$ $n_{\mathcal{B}} = n_{\mathcal{I}} = 125$ | $n_{\mathcal{N}} = 10000$ $n_{\mathcal{B}} = n_{\mathcal{I}} = 250$ | $n_{\mathcal{N}} = 20000$ $n_{\mathcal{B}} = n_{\mathcal{I}} = 500$ |
|---|---|---|---|
| Adam | $1.494 \pm 0.060$ | $1.484 \pm 0.061$ | $1.438 \pm 0.031$ |
| PCGrad | $1.350 \pm 0.015$ | $1.344 \pm 0.019$ | $1.331 \pm 0.037$ |
| IMTL-G | $1.368 \pm 0.012$ | $1.339 \pm 0.024$ | $1.275 \pm 0.003$ |
| MinMax | $1.943 \pm 0.212$ | $1.889 \pm 0.143$ | $1.846 \pm 0.135$ |
| ReLoBRaLo | $1.478 \pm 0.067$ | $1.419 \pm 0.053$ | $1.381 \pm 0.043$ |
| LRA | $340.161 \pm 135.263$ | $353.796 \pm 114.972$ | $374.574 \pm 84.340$ |
| ConFIG | $1.307 \pm 0.014$ | $1.308 \pm 0.008$ | $1.295 \pm 0.014$ |
| M-ConFIG | $\mathbf{1.291 \pm 0.017}$ | $\mathbf{1.277 \pm 0.035}$ | $\mathbf{1.262 \pm 0.004}$ |

Table 27: The performance of different methods with different ratios of data samples during training

|  | $n_{\mathcal{N}} = 5000$ $n_{\mathcal{B}} = n_{\mathcal{I}} = 250$ | $n_{\mathcal{N}} = 10000$ $n_{\mathcal{B}} = n_{\mathcal{I}} = 250$ | $n_{\mathcal{N}} = 20000$ $n_{\mathcal{B}} = n_{\mathcal{I}} = 250$ |
|---|---|---|---|
| Adam | $1.470 \pm 0.012$ | $1.484 \pm 0.061$ | $1.455 \pm 0.044$ |
| PCGrad | $1.358 \pm 0.026$ | $1.344 \pm 0.019$ | $1.355 \pm 0.013$ |
| IMTL-G | $1.328 \pm 0.036$ | $1.339 \pm 0.024$ | $1.342 \pm 0.027$ |
| MinMax | $1.963 \pm 0.133$ | $1.889 \pm 0.143$ | $1.843 \pm 0.152$ |
| ReLoBRaLo | $1.469 \pm 0.014$ | $1.419 \pm 0.053$ | $1.393 \pm 0.052$ |
| LRA | $362.388 \pm 110.455$ | $353.796 \pm 114.972$ | $350.266 \pm 127.639$ |
| ConFIG | $1.298 \pm 0.014$ | $1.308 \pm 0.008$ | $1.300 \pm 0.010$ |
| M-ConFIG | $\mathbf{1.289 \pm 0.015}$ | $\mathbf{1.277 \pm 0.035}$ | $\mathbf{1.265 \pm 0.026}$ |

