# OpenReview forum: "ConFIG: Towards Conflict-free Training of Physics Informed Neural Networks"
_ICLR.cc/2025/Conference — ICLR 2025 Spotlight_

### Official Review · Reviewer_4Pd7 · 2024-10-28

**Soundness:** 3
**Presentation:** 3
**Contribution:** 3
**Rating:** 6
**Confidence:** 4

**Summary:**

The paper introduces a novel method, ConFIG (Conflict-Free Inverse Gradients), aimed at addressing gradient conflicts that arise during the training of Physics-Informed Neural Networks (PINNs). The primary challenge in training PINNs lies in the conflicting gradient directions from multiple loss terms, particularly those associated with initial/boundary conditions and the physics equations, which can impede learning and optimization.  The paper proposes the ConFIG method, which ensures that the final update gradient has a positive dot product with each loss-specific gradient, eliminating conflicts and enabling consistent optimization across all loss terms. The method adjusts gradient magnitudes dynamically based on conflict levels and maintains uniform optimization rates for all loss terms. The method provides a robust and conflict-free optimization framework for PINNs, and its application extends to multi-task learning, showcasing improved performance over existing methods.

**Strengths:**

The paper makes a notable contribution to the field of PINN training by addressing a fundamental challenge with an original and technically sound approach.

Originality: The ConFIG method introduces a novel solution to gradient conflict resolution in PINNs. It effectively adapts ideas from multi-task learning (MTL) and continual learning (CL) to address gradient conflicts, which is relatively unexplored in the PINN domain.

Quality: The paper demonstrates strong technical quality with rigorous mathematical proofs and comprehensive experimental validation across multiple benchmark problems. While the method’s effectiveness is well-supported, additional comparisons with state-of-the-art methods and scalability analysis would further bolster its quality.

Clarity: The paper is generally well-organized and clear, though certain sections, particularly those with heavy mathematical notation, could be simplified for broader accessibility.

Significance: The method addresses a critical challenge in PINN training, with broad implications for fields like fluid dynamics, electromagnetics, and beyond.

**Weaknesses:**

The paper would significantly increase its rigor and relevance by including more comprehensive comparisons, and the detailed comments are
1. Limited Benchmarking: The evaluation is narrow, focusing on standard PDEs like Burgers equations, without addressing more complex, real-world PDE challenges (e.g., multi-scale phenomena, high-dimensionality). It would be helpful if the authors could broaden the evaluation to include more diverse and complex PDEs, as seen in benchmarks like PINNacle (Hao et al., 2024), to demonstrate the robustness of ConFIG across a wider range of problems.

2. Comparison with Advanced Methods: The paper primarily compares ConFIG to basic methods (e.g., PCGrad, IMTL-G), without considering recent, more sophisticated approaches in PINN optimization. The authors are expect to include comparisons with advanced methods.

3. Scalability and Efficiency: The scalability of ConFIG, especially regarding the computational cost is not fully explored. It would be better if the authors could provide a detailed analysis of runtime and memory efficiency, particularly for large-scale or high-dimensional problems, to validate ConFIG’s scalability.

4. Ablation and Sensitivity Studies: There’s a lack of ablation studies to assess the impact of key hyperparameters like learning rate and gradient scaling. The authors could perform more ablation studies to better understand the effect of different components and configurations on ConFIG’s performance.

**Questions:**

1. What is the computational complexity of the pseudoinverse operation in ConFIG, and how does it scale with larger problem sizes?

2. It would be better if the authors could test ConFIG on high-dimensional or multi-scale PDE problems, as seen in benchmarks like PINNacle? If so, how does it perform?

3. The authors are expected to provide more insight into how sensitive ConFIG is to hyperparameters like learning rate and batch size?

4. The authors are expected to evaluat ConFIG’s scalability and efficiency for large-scale industrial applications.

---

> ### Author Response · Authors · 2024-11-21
>
> We would like to thank the reviewer for the detailed review and feedback. In the following section, we will address the weaknesses (W) and questions (Q) mentioned above.
>
>
>
>
>
>
>
> - **W1&Q2** *" It would be helpful if the authors could broaden the evaluation to include more diverse and complex PDEs, as seen in benchmarks like PINNacle (Hao et al., 2024)."*; "It would be better if the authors could test ConFIG on high-dimensional or multi-scale PDE problems, as seen in benchmarks like PINNacle":
>
>
>
>   We agree that more benchmark tests help demonstrate the ability of our method. In the revised version of the paper, we provide three new tests following the PINNacle benchmark configuration:
>
>
>
>     - **High-dimensional case**: We tested a 5-dimensional Poisson's equation. The results show that M-ConFIG outperforms all benchmark methods, with a $76$% improvement relative to Adam. The ConFIG method still yields an improvement of $73$%.
>
>
>
>
>
>   - **Multi-scale case:** We evaluated a multi-scale heat transfer problem. Although all other benchmark methods struggled, M-ConFIG achieved the best performance with an average improvement of $14$%, while ConFIG showed a $10$% improvement.
>
>
>
>
>
>   - **Chaotic case**: We analyzed the Kuramoto-Sivashinsky (KS) equation as a representative chaotic PDE. As reported in PINNacle, all methods, including ours, failed to accurtately capture the transition to chaotic behavior.
>
>
>
>
>
> ​	All these tests show that our ConFIG and M-ConFIG methods do improve the result in most cases. At the same time, we also need to be aware that training PINNs is a challenging topic and that some issues, e.g., addressing the chaotic regime, are not induced by the conflict of different loss terms. These challenges require improvement from other sides, like improved neural network architecture.
>
>
>
>
>
>
>
> - **W2** " *The paper primarily compares ConFIG to basic methods (e.g., PCGrad, IMTL-G), without considering recent, more sophisticated approaches in PINN optimization. The authors are expect to include comparisons with advanced methods*":
>
>
>
>   We are sincerely surprised about this comment. Unfortunately, it also lacks any mention of specific _advanced methods_ that were omitted. To the best of our knowledge, our comparisons cover all commonly used basic and advanced methods for PINN training. For brevity, we actually did not include two additional methods in the main text:
>
>
>
>   - We tested the Multi-Adam method (Yao et al., 2023), and it shows **negative** performance for all tests in our paper and the new PINNacle cases. Thus, we didn't include it in our manuscript.
>
>   - We also tested the NTK method (Wang et al., 2020), which requires the evaluation of the Neural Tangent Kernel to determine the weights of loss terms. This involves calculating the full Jacobian matrix of the loss terms, and is computationally **very expensive**. The GPU memory and computational time costs for calculating the Jacobi matrix are still 1GB and 3.8s in a single iteration for the simplest 1D Burgers equation. This does not include the cost of the training procedure yet. For all other methods, the total memory will not exceed 400 MB, and the average training speed is around 100 iterations/seconds. This makes the NTK method impractical.
>
>
>
>   Meanwhile, although PCGrad and IMTL-G methods are well established in the MTL community, their performance is **not well investigated in the PINN tests**. To our knowledge, the only published work that involves gradient-based PINN tests is from Zhou et al. (A generic physics-informed neural network-based framework for reliability assessment of multi-state systems, 2023). They use the PCGrad method to balance the PINN's training for multi-state systems. Our manuscript is the first to evaluate the performance of these gradient-based methods among different standard PINN tests. Thus, we do not consider PCGrad and IMTL-G methods as "common ground" for the PINN community. We believe that bringing some of the MTL advances into the PINN community is one of the contributions of the current study.

---

> > ### Author Response · Authors · 2024-11-21
> >
> > - **W3&Q1&Q4** “It would be better if the authors could provide a detailed analysis of runtime and memory efficiency”；“What is the computational complexity of the pseudoinverse operation in ConFIG, and how does it scale with larger problem sizes”; "The authors are expected to evaluat ConFIG’s scalability and efficiency for large-scale industrial applications":
> >
> >
> >
> >   We agree with you that scalability and efficiency are important for a method, especially when dealing with large-scale problems. To answer your questions about performance and sensitivity:
> >
> >
> >
> >   - **Computational complexity**: The ConFIG method scales **linearly with the dimensionality of the gradient (the number of trainable parameters) and super-linearly in the number of gradients**. We also show experimentally that the pseudo inverse cost is insignificant compared to the cost of loss backpropagation. Details of the computational complexity are provided Appendix A.6.
> >
> >
> >
> >   - **Run time efficiency**: In the original manuscript, we discussed run time efficiency in the last paragraph of Section 3.3. We introduce a coefficient to investigate the training efficiency of ConFIG and M-ConFIG. We showed that gradient-based methods, including IMTL-G, PCGrad, and our ConFIG method, are usually slower than the loss weighting methods and the original Adam optimizer. However, with the help of momentum acceleration, we can achieve even **faster training efficiency than the standard Adam method**.
> >
> >
> >
> >   - **Memory efficiency**:  The gradient-based methods, including PCGrad, IMTL-G, and our ConFIG method, usually require more memory compared to the weighting methods during training since they need to store the gradient for each loss term in each iteration. Meanwhile, while the momentum acceleration helps to increase the training speed, it also requires more memory as it needs to store the first and second momentum for each gradient during the training procedure. For both the gradient-based method with and without momentum acceleration, the complexity of the additional memory is $\mathcal{O}(nm)$, where n is the number of loss terms and m is the number of trainable parameters.
> >
> >
> >
> >   We didn't discuss memory efficiency in the original version of the manuscript, thank you for pointing out this omission. In the revised version, **a discussion will be added in Appendix A.6**.
> >
> >
> >
> >
> >
> >
> >
> > - **W4&Q3** *"The authors could perform more ablation studies to better understand the effect of different components and configurations on ConFIG’s performance." "The authors are expected to provide more insight into how sensitive ConFIG is to hyperparameters like learning rate and batch size?"*:
> >
> >
> >
> >   In Appendix A.13 of the revised manuscript, we have added an ablation study for the learning rate, batch size, and ratio between sample points. The results show that **the superiority of our method does not change with different training parameters**, which further shows the robustness of the proposed method.
> >
> >
> >
> > We hope that these explanations address your concerns, but we'd be happy to answer any remaining questions about our method. We will upload the revised version of the manuscript shortly.

---

> > > ### Comment · Reviewer_4Pd7 · 2024-11-23
> > >
> > > Thanks for your response and I raised my score.

---

### Official Review · Reviewer_PVNV · 2024-11-03

**Soundness:** 3
**Presentation:** 4
**Contribution:** 3
**Rating:** 8
**Confidence:** 3

**Summary:**

This paper addresses the challenge of gradient conflicts in Physics-Informed Neural Networks (PINNs) and the tendency of optimization to get stuck in the local minima of specific loss terms. The authors highlight that the gradient magnitude of PDE residuals typically dominates that of initial and boundary condition losses, leading to biased final updates. Their proposed ConFIG method leverages a momentum-based approach that reduces computational costs by alternating the backpropagation of loss-specific gradients, ensuring conflict-free updates and improved convergence.

**Strengths:**

- The optimization of different loss functions in PINNs is critical, as it can influence whether PINNs succeed or fail in solving forward problems. The exploration of Multi-Task Learning (MTL) strategies for improving PINNs training is innovative and may inspire future methodologies.
- The problem setup and method are clearly explained, making the paper accessible and informative. The theoretical background is well-supported by proofs and discussions.
- The experimental design is good, but it needs some adjustments.
- Given the difficulty of training PINNs due to conflicting gradient directions and numerical issues, the ConFIG method can be a good direction for solving these problems.

**Weaknesses:**

- The examples explored in this study are relatively simple and can be solved with standard methods like Adam. It would be insightful to evaluate the proposed approach on more complex, chaotic PDEs, such as the Kuramoto-Sivashinsky equation, which poses significant challenges due to its non-linear and chaotic behavior.
- The study omits comparisons with certain MTL strategies, such as FAMO, which could provide further context to the method’s effectiveness in PINNs. Additionally, using only the CelebA dataset to demonstrate performance in MTL may not be comprehensive enough.

**Questions:**

In addition to what I mentioned in weaknesses, I would like to know how this type of models can be compared to traditional methods to improve PINN training typically involves adjusting the weights for PDE residuals and loss terms for initial and boundary conditions in terms of convergence and performance.

---

> ### Author Response · Authors · 2024-11-21
>
> We would like to thank the reviewer for the detailed review and feedback. In the following section, we will address the weaknesses (W) and questions (Q) mentioned above.
>
>
>
>
>
>
>
> - **W1** *"It would be insightful to evaluate the proposed approach on more complex, chaotic PDEs, such as the Kuramoto-Sivashinsky equation"*:
>
>
>
>   We agree with the reviewer that more challenging tests will help demonstrate the ability of our method. In the revised version of the paper, we provide three new tests following the PINNacle benchmark (Hao et al., 2023) configuration:
>
>
>
>   - **High-dimensional case**: We tested a 5-dimensional Poisson's equation. The results show that M-ConFIG outperforms all benchmark methods, with a $76$% improvement relative to Adam. The ConFIG method still yields an improvement of $73$%.
>
>
>
>   - **Multi-scale case:** We evaluated a multi-scale heat transfer problem. Although all other benchmark methods struggled, M-ConFIG achieved the best performance with an average improvement of $14$%, while ConFIG showed a $10$% improvement.
>
>
>
>   - **Chaotic case**: We analyzed the Kuramoto-Sivashinsky (KS) equation as a representative chaotic PDE. As reported in PINNacle, all methods, including ours, failed to accurtately capture the transition to chaotic behavior.
>
>
>
> These additional tests show that our ConFIG and M-ConFIG methods do improve the results for all cases that are solvable with Adam-like optimizers. At the same time, we also need to be aware that training PINNs is a challenging topic and that some issues, e.g., addressing the chaotic regime, are not induced by the conflict of different loss terms. These challenges require improvement from other sides, like improved neural network architecture.
>
>
>
>
>
>
>
> - **W2** *"The study omits comparisons with certain MTL strategies, such as FAMO"; "Using only the CelebA dataset to demonstrate performance in MTL may not be comprehensive enough"*:
>
>   - We agree that integrating more benchmark methods from MTL research into the PINNs test is helpful to enhance the quality of the manuscript. In the current study, we only involve the PCGrad and IMTL-G methods from the MTL study to the PINN test. On the one hand, IMTL-G and PCGrad are similar to our method as they both utilize the gradient information. On the other hand, these methods are also the only ones previously used for PINNs studies (A generic physics-informed neural network-based framework for reliability assessment of multi-state systems, 2023). Other methods, including FAMO, did not perform better than our method in our MTL test. Correspondingly, we focused our evaluation of PINN scenarios on methods from that area.
>
>
>
>   - We also tested the FAMO method on the burgers equation during the previous days. We directly use the implementation from the official FAMO repo and integrated it into our test code. **The result shows that FAMO failed to train the PINNs for burgers**. The average test loss for the two-loss scenario and the three-loss scenario are 0.368 and 0.289, respectively. By examining the loss curve, we find the FAMO method gets stuck on optimizing the PDE residual and ignoring the contribution from the boundary and initial conditions. Nonetheless, we agree that comparing more methods from the MTL community will be an interesting and important topic for the PINN community.
>
>
>
>   - Although the current study mainly focuses on addressing the conflict issues during the training of PINNs, we agree that investigating the proposed method's performance in more MTL experiments is a nice idea. In fact, as we discussed in the introduction section, many other methods need to address the conflict during the training, e.g., continuous learning. Comprehensively investigating the proposed method in these applications and improving the methods correspondingly would be a very exciting direction in the future.

---

> > ### Author Response · Authors · 2024-11-21
> >
> > - **Q1** *" I would like to know how this type of models can be compared to traditional methods to improve PINN training typically involves adjusting the weights for PDE residuals and loss terms for initial and boundary conditions in terms of convergence and performance."*:
> >
> >
> >
> >   The traditional weighting method in PINNs only considers the information from scalar losses, while our method considers the information from the gradient vector of each loss. **The loss-based weighting method can't fully resolve the conflict during the training**, as changing the weights of losses only changes the magnitude of the corresponding optimization gradient. Our gradient-based method can change not only the magnitude of each loss-specific gradient but also their directions. In our PINN tests, the LRA, MinMax, and ReLoBRaLo methods all belong to the family of weighting methods. Our results show that **the proposed method consistently performs better than these weighting methods.** Meanwhile, it is also possible to integrate the weighting methods into our gradient-based framework. This can be done by using the weights to set the direction bias of the update gradient, as outlined in the "Adjusting direction weights" paragraph of our submission. However, for our tests, adjusting the weights did not show additional benefits.
> >
> >
> > We hope that these explanations address the reviewer's remaining concerns, but we'd be happy to answer any remaining questions about our method. We will upload the revised version of the manuscript shortly.

---

> > > ### Comment · Reviewer_PVNV · 2024-11-25
> > >
> > > Thank you for your detailed response. It was interesting to learn that these approaches cannot solve chaotic problems. Including the solution and prediction figure could help illustrate the complexity and difficulty of such problems. After considering the other reviews, I have decided to increase my initial score.

---

### Official Review · Reviewer_Gt6s · 2024-11-03

**Soundness:** 4
**Presentation:** 3
**Contribution:** 4
**Rating:** 8
**Confidence:** 4

**Summary:**

This paper proposes a novel method (ConFIG) to deal with multi-task problems, and specifically focus on the task of training physics-informed neural networks (PINNs). The main motivation behind the method is that multiple loss functions in their formulation (PDE residuals, boundary / initial conditions) often conflict and lead to suboptimal optimizations. To overcome this limitations, ConFIG ensures that the update direction is conflict free by maintaining a positive-dot product between the final direction and the individual loss-specific gradients. Additionally, the paper introduces a momentum-based version (M-ConFIG) which reduces computational efficiency while maintaining training efficiency. The proposed method is contrasted against other loss aggregation schemes on several PDE problems and a multi-task learning benchmark, and demonstrates improved performance over existing methods.

**Strengths:**

- The paper addresses the important issue of conflicting directions in gradient updates in the training of PINNs. The proposed approach is novel and seems to combine the best of other MTL methods that have been applied in the context of PINNs such as PCGrad and IMTL-G. The proposed formulation seems well-formulated and is theoretically sound. Further, the introduction of M-ConFIG as a method to accelerate training while avoiding the evaluation of all the individual gradients at every step is a great practical contribution. Additionally, the appendices on this aspect are also thorough and discuss the reason as to why using Adam with ConFIG wouldn't work.

- The experiments also reveal a better performance than other baselines across different PDE benchmarks. Moreover, the method also performs well when applied to the standard MTL CelebA benchmark revealing broader applicability. The experiments also showed better convergence when compared against computational time of other optimization schemes further improving the applicability.

**Weaknesses:**

- The experiments that are peformed in the paper compare against other MTL aggregation schemes for the proposed benchmark problems. However, there are several challenging benchmark problems for PINNs that have been established in other works (such as PirateNets (Wang et al.), PINNacle (Hao et al.),  Expert's Guide to PINNs (Wang et al.)). Although, they do use other enhancements such as an improved network architecture / improved encodings / imposing causality, it would be interesting to see if these enhancements would further improve on the current state of the art results.

- The hyperparameter tuning and ablation studies in the paper could be more extensive. For instance, the authors claim that the learning rate was fixed to $10^{-4}$ for the long-training or the use of a $10^{-3} \to 10^{-4}$ decay. But they don't seem to study the influence of this chosen rate on the final relative L2 error that has been obtained. It would strengthen the position of the paper to study this aspect as well.

**Questions:**

In addition to the points raised in weaknesses, it would be good if the authors can address the following:
- It seems like this approach has been primarily studied in the context of forward problems. Have you performed any studies on inverse problems as well?
- Additionally, I'm curious how the more recent pre-print of Jacobian Descent for Multi-Objective Optimization (Quinton & Rey) ties in, and compares against your work. Again, they compare against several MTL aggregators and propose a new strategy that claims to avoid conflicts with any of the individual gradients.

---

> ### Author Response · Authors · 2024-11-21
>
> We would like to thank the reviewer for the detailed review and feedback. In the following section, we will address the weaknesses (W) and questions (Q) mentioned above.
>
>
>
>
>
>
>
> * **W1** _"However, there are several challenging benchmark problems for PINNs that have been established in other works (such as PirateNets (Wang et al.), PINNacle (Hao et al.), Expert's Guide to PINNs (Wang et al.))."_:
>
>
>
>    We appreciate the suggestion to incorporate more benchmarks. In the revised manuscript, we include the following new tests based on the PINNacle benchmark:(Hao et al., 2023):
>
>
>
>   - **High-dimensional case**: We tested a 5-dimensional Poisson's equation. The results show that M-ConFIG outperforms all benchmark methods, with a $76$% improvement relative to Adam. The ConFIG method still yields an improvement of $73$%.
>
>
>
>   - **Multi-scale case:** We evaluated a multi-scale heat transfer problem. Although all other benchmark methods struggled, M-ConFIG achieved the best performance with an average improvement of $14$%, while ConFIG showed a $10$% improvement.
>
>
>
>   - **Chaotic case**: We analyzed the Kuramoto-Sivashinsky (KS) equation as a representative chaotic PDE. As reported in PINNacle, all methods, including ours, failed to accurtately capture the transition to chaotic behavior.
>
>
>
> ​	All these tests show that our ConFIG and M-ConFIG methods do improve the result in most cases. At the same time, we also need to be aware that training PINNs is a challenging topic and that some issues , e.g., addressing the chaotic regime, are not induced by the conflict of different loss terms.  As the reviewer mentioned, integrating other technology like improved network architecture, improved encodings and  imposing causality will be a very interesting future research direction.
>
>
>
>
>
>
>
> * **W2** _"The hyperparameter tuning and ablation studies in the paper could be more extensive."_:
>
>
>
>   To address hyperparameter sensitivity, we have added an ablation study in Appendix A.13 of the revised manuscript. We examine the effects of learning rate, batch size, and sample ratios, confirming that **our method's superiority holds across varied configurations**. This reinforces the robustness of our approach.

---

> > ### Author Response · Authors · 2024-11-21
> >
> > * **Q1** _"Have you performed any studies on inverse problems as well?"_:
> >
> >
> >
> >   We have not yet explored inverse problems, but this is a promising area for future work. Given the consistent performance improvements in forward problems, we anticipate similar benefits in inverse problems. Additionally, the inherent **conflicts between PDE losses and observational data** in inverse problems present an ideal use case for our method.
> >
> >
> >
> >
> >
> >
> >
> > * **Q2** _"I'm curious how the more recent pre-print of Jacobian Descent for Multi-Objective Optimization (Quinton & Rey) ties in"_:
> >
> >
> >
> >   Thank you for bringing this preprint to our attention, we have integrated this paper into the related work discussion in the revised manuscript.
> >
> >
> >
> >   * In this paper, they proposed a Jacobian Descent method as an alternative to the gradient descent optimization. They concatenate multiple loss terms as a vector and calculate the Jacobian of this loss vector w.r.t optimization parameters. They also introduce an "aggregator" to reshape the Jacobian matrix as a row vector to update the optimization parameters. This basic configuration is mathematically the same as the "gradient-based" methods, e.g., the IMTL-G, PCGrad, and our ConFIG method, discussed in our paper. The loss-specific gradients in gradient-based methods are actually row vectors of the Jacobian matrix in Jacobian Descent. Gradient-based methods then calculate the final update gradient with these row vectors, corresponding to the "aggregator" in the context of "Jacobian Descent." For this initial formulation, **the Jacobian Descent and previous gradient-based methods are equivalent**.
> >
> >
> >
> >
> >
> >   - A contribution of the "Jacobian Descent for Multi-Objective Optimization" method is a new aggregator "UPGrad". Like the PCGrad method, the UPGrad method projects the loss-specific gradients onto the boundary of **the positive cone,** an area in optimization parameter space where the gradients inside this cone have a positive dot product to each loss-specific gradient. Our ConFIG method also guarantees that the final update gradient is inside the positive cone. The major difference between our ConFIG method and their UPGrad method is that **our ConFIG method makes sure that the final update gradient is in the center of the positive cone** due to the equal direction weights. In the UPGrad method, although the final update gradient is inside the cone, it still **biases some of the loss-specific gradients**. According to our test in the "Adjusting direction weights" paragraph, the equal weight configuration will give better performance. Nonetheless, comprehensive experiments are still needed to evaluate the performance of different methods. While the UPGrad and our ConFIG methods use different ways to guarantee the final update gradient is inside the cone, it is also worth comparing the efficiency of the two methods in the future. The UPGrad paper does not propose modifications like our M-ConFIG method, which we expect to outperform UPGrad in terms of runtime.
> >
> >
> >
> >
> >
> >
> >
> > We hope that these explanations address the reviewer's remaining concerns, but we'd be happy to answer any remaining questions about our method. We will upload the revised version of the manuscript with the new results shortly.

---

> > > ### Comment · Reviewer_Gt6s · 2024-11-28
> > > **Questions Addressed**
> > >
> > > I appreciate the authors' thoughtful efforts in addressing my questions and concerns. I am glad to see that my concerns have been addressed with additional experiments to demonstrate the strengths of the proposed method. I will maintain my original score and continue to recommend your paper for acceptance. Thank you for your detailed responses and clarifications!

---

### Official Review · Reviewer_5ieJ · 2024-11-04

**Soundness:** 4
**Presentation:** 4
**Contribution:** 3
**Rating:** 8
**Confidence:** 4

**Summary:**

Motivated by the discrepancy between the residual and data loss terms in Physics-Informed Neural Networks (PINNs), the authors propose a "conflict-free" gradient update for multi-task learning, such that the resulting optimization direction minimizes all individual loss terms, avoiding local minima induced by one loss term. Such optimization step is calculated by ensuring positive cosine similarity between the update direction and the individual gradients of each loss term and (in most cases) a uniform decay rate for all losses. The authors also propose a momentum-based algorithm to alleviate the computational costs.

**Strengths:**

1.  The experiments thoroughly evaluate the proposed method from various aspects and for different problems.
    - The effectiveness (in terms of accuracy) of both ConFIG and Momentum(M)-ConFIG algorithms is demonstrated compared to other gradient adjustment methods, loss weighting methods for PINNs, and Adam.
    - While M-ConFIG falls short of the original ConFIG, its performance in computation-constrained scenarios is well shown.
2. The motivating problem from PINNs is a significant obstacle to wider adoption of them, and the proposed algorithm paves the way for their application to more problems in Physics.
3. The paper is well-written and accompanied by a reasonable amount of experiments, maths, and pseudo-code.

**Weaknesses:**

1. Although the experiments are already extensive, the paper can benefit from experiments with other PDEs (e.g. KS, Turbulence) and boundary conditions (e.g. Neumann) from the studied loss weighting baselines.

**Questions:**

NA

---

> ### Author Response · Authors · 2024-11-21
>
> We thank the reviewer for their suggestions. We fully agree that additional challenging tests would further strengthen our study. In the revised version of the manuscript, we have included three new experiments based on the PINNacle benchmark (Hao et al., 2023) configuration:
>
>
>
>
>
>
>
> - **High-dimensional case**: We tested a 5-dimensional Poisson's equation. The results show that M-ConFIG outperforms all benchmark methods, with a $76$% improvement relative to Adam. The ConFIG method still yields an improvement of $73$%.
>
>
>
> - **Multi-scale case:** We evaluated a multi-scale heat transfer problem. Although all other benchmark methods struggled, M-ConFIG achieved the best performance with an average improvement of $14$%, while ConFIG showed a $10$% improvement.
>
>
>
> - **Chaotic case**: We analyzed the Kuramoto-Sivashinsky (KS) equation as the representative chaotic PDE. As reported in PINNacle, all methods, including ours, failed to accurtately capture the transition to chaotic behavior.
>
>
>
>
>
>
>
> All these tests show that our ConFIG and M-ConFIG methods do improve the result in most cases. At the same time, we also need to be aware that training PINNs is a challenging topic and that some issues are not induced by the conflict of different loss terms. The KS case serves as an example to highlight the outstanding challenges. They require other modifications, potentially via improved network architectures, improved encodings and imposed causality.  We also recognize the importance of exploring other challenging PDEs, such as turbulence and those with complex boundary conditions, as potential future research directions. We hope these clarifications address the reviewer's concerns, and we remain open to further questions. The revised manuscript will be uploaded shortly, and will be expanded with a discussion of the new test cases.

---

> > ### Comment · Reviewer_5ieJ · 2024-11-28
> >
> > Thanks for your response. Considering your rebuttal and other reviews as well, I continue to support the paper and maintain my initial score with a higher confidence score.

---

### Author Response · Authors · 2024-11-21
**General Reply to All Reviewers**

Dear Reviewers,







We would like to express our gratitude for your insightful feedback. Building on your comments and our rebuttal discussion, we have performed additional experiments that we'd like to share:







* We test three new challenging PINN cases, following your suggestions, from the PINNacle benchmark (Hao et al., 2023). The results are shown in **Appendix A.11.** Although, like all other benchmarked methods, our method is not able to handle some of the extremely challenging cases, the proposed ConFIG and M-ConFIG methods outperform the other methods. The gains are especially prominent in the high-dimensional case.







* We perform an additional ablation study on the training hyperparameters. The results are shown in **Appendix A.13.** The results show that the superiority of (M-)ConFIG persists over a wide range of different training parameters. This further demonstrates the robustness of our method.







These new results can now be found in the updated PDF. We also introduced other modifications according to the requirements of your reviews; you can find a detailed discussion in the individual replies. All the modifications have been highlighted in
magenta.







Once again, we thank all reviewers for their detailed and constructive feedback. Your comments help to improve future revisions of our work.

---

### Meta-Review · Area_Chair_2D7Z · 2024-12-20

**Metareview:**

This paper proposes **ConFIG** for multi-objective optimization, ensuring positive cosine similarity between update directions and consistent optimization rates for all loss terms. The authors also introduce a momentum-based variant to accelerate convergence. The paper presents extensive experiments, focusing in particular on physics-informed neural networks (PINNs), but also demonstrating improvements on a general multi-task benchmark. All reviewers gave very positive feedback. Although the approach still cannot solve certain challenging chaotic PDE problems, this requires innovations beyond the scope of this work. Given strong reviews and positive results across extensive experiments, I recommend accepting this paper as spotlight.

**Additional Comments On Reviewer Discussion:**

Initially, there were general concerns about the limited PINN benchmarks (Reviewers 5ieJ, Gt6s, PVNV, 4Pd7), the restricted set of baselines (Gt6s, PVNV, 4Pd7), and potential hyperparameter sensitivity (Gt6s, 4Pd7). Following the authors’ rebuttal, these issues were largely resolved, as the authors provided extended discussions of related work and additional experimental results.

---

### Decision · Program_Chairs · 2025-01-22

Accept (Spotlight)